# TASK-AGNOSTIC FEDERATED CONTINUAL LEARNING VIA REPLAY-FREE GRADIENT PROJECTION

## ABSTRACT

Federated continual learning (FCL) enables distributed client devices to learn from streaming data across diverse and evolving tasks. A major challenge to continual learning, catastrophic forgetting, is exacerbated in decentralized settings by the data heterogeneity, constrained communication and privacy concerns. We propose *Federated gradient Projection-based Continual Learning with Task Identity Prediction* (**FedProTIP**), a novel FCL framework that mitigates forgetting by projecting client updates onto the orthogonal complement of the subspace spanned by previously learned representations of the global model. This projection reduces interference with earlier tasks and preserves performance across the task sequence. To further address the challenge of task-agnostic inference, we incorporate a lightweight mechanism that leverages core bases from prior tasks to predict task identity and dynamically adjust the global model's outputs. Extensive experiments across standard FCL benchmarks demonstrate that FedProTIP significantly outperforms state-of-the-art methods in average accuracy, particularly in settings where task identities are a priori unknown. Our code is available here.

## 1 INTRODUCTION

Federated learning (FL) (McMahan et al., 2017), where client devices collaboratively train a global model without sharing private data, has emerged as a compelling alternative to centralized learning. Most FL systems assume static local datasets and a single inference task per client. In practice, however, devices (e.g., phones, smart glasses) often collect data for multiple evolving tasks and must fine-tune the model over time. Limited storage further forces clients to discard old data, creating a continual learning (CL) scenario where models must adapt to new tasks without having access to prior ones. This setting exacerbates *catastrophic forgetting* (McCloskey & Cohen, 1989), i.e., deterioration of the model's performance on previously learned tasks. In federated systems that operate under continual learning, this problem is further intensified by the data heterogeneity across client devices.

Many studies have explored adapting conventional continual learning schemes to federated settings, typically falling into three categories: (1) *replay-based* methods (Dong et al., 2022; Liu et al., 2023; Dai et al., 2023; Li et al., 2024c;a); (2) *generation-based* methods (Qi et al., 2023; Zhang et al., 2023; Tran et al., 2024; Liang et al., 2024; Yu et al., 2024); and (3) *regularization-based* methods (Yoon et al., 2021; Ma et al., 2022; Li et al., 2024b; Lee et al., 2024). Each faces challenges in FL: replay methods store old examples, risking privacy or exceeding storage limits; generation methods require server-side generative models, slowing aggregation; and regularization methods often impose significant local compute overhead. Recently, gradient projection methods such as GPM (Saha et al., 2021) have proven effective in centralized continual learning by projecting new task gradients onto subspaces orthogonal to those of previously learned tasks. However, GPM assumes centralized access to activation statistics, which renders it impractical in federated settings. While FOT (Bakman et al., 2024) adapts GPM to FL by collecting high-dimensional embeddings from clients, this approach incurs significant communication overhead and raises potential privacy concerns.

In this work, we propose Federated gradient Projection-based continual learning with Task Identity Prediction (**FedProTIP**), a novel federated continual learning (FCL) framework that combines subspace-based gradient projection with inference-time task prediction to address both forgetting and task ambiguity. FedProTIP mitigates catastrophic forgetting by projecting local gradients onto the subspace orthogonal to that spanned by previously learned features; this reduces interference between representations across tasks. Specifically, each client collects layer-wise activations at the end of local training on its current task and performs randomized singular value decomposition (SVD) on the collected activations to extract the core bases of the task's feature subspace. These

local core bases are sent to the server, which constructs a projection matrix and broadcasts it back to the clients. The clients then perform projected gradient descent to fine-tune their local models – a strategy that preserves earlier features by ensuring updates are orthogonal to prior task subspaces. Beyond gradient projection, FedProTIP introduces a novel *task identity prediction* mechanism that exploits the extracted gradient subspaces to estimate task relevance during inference. By leveraging these subspaces to infer the most likely task identity of each test input, FedProTIP dynamically adjusts the global model's outputs, yielding significant performance improvements in the challenging task-agnostic federated continual learning setting. Extensive experimental results demonstrate that FedProTIP outperforms existing FCL approaches across a range of benchmarking datasets.

The main contributions of this paper can be summarized as follows:

- We introduce *FedProTIP*, a novel federated continual learning framework that mitigates catastrophic forgetting through subspace-based gradient projection. Unlike prior approaches, FedProTIP performs local gradient projection and communicates only compact core subspace bases to the server, preserving data privacy and achieving efficiency with minimal computational and memory overhead.

- We develop a novel *task identity prediction* method that leverages gradient subspace alignment to infer task-ID at inference and dynamically route inputs to the appropriate output heads. This removes the unrealistic assumption of known task identities and delivers strong gains in task-agnostic federated continual learning, without relying on replay buffers, generative models, or auxiliary prediction modules.

- We conduct extensive experiments on multiple continual image classification benchmarks, showing that FedProTIP consistently outperforms state-of-the-art FCL methods in both accuracy and forgetting, with accuracy improvements ranging from at least 4.3% up to 47%.

## 2 RELATED WORK

### 2.1 FEDERATED CONTINUAL LEARNING

Federated continual learning (FCL) tackles the challenge of continuously learning from decentralized data while preserving performance across sequential tasks. An early FCL approach, FedWeIT (Yoon et al., 2021), decomposes model parameters into task-generic and task-specific components, focusing on a task-incremental setting where the task ID is known during inference. CFeD (Ma et al., 2022) relies on knowledge distillation using a surrogate dataset shared between the server and clients. GLFC (Dong et al., 2022; 2023) addresses catastrophic forgetting by combining class-aware gradient compensation with class-semantic relation distillation, but relies on storing examples from previous tasks. Subsequent works (Liu et al., 2023; Dai et al., 2023; Li et al., 2024c;a) reduce the size of the replay cache, yet remain reliant on stored samples.

Recently, several FCL methods have leveraged generative models to replace real examples in memory with synthetic data. FedCIL (Qi et al., 2023) employs a GAN with an auxiliary classifier to enable generative replay, thereby mitigating forgetting while aggregating global knowledge across clients. TARGET (Zhang et al., 2023) and MFCL (Babakniya et al., 2024) introduce data-free knowledge distillation that uses synthetic examples to transfer knowledge from a previously trained global model to client models. LANDER (Tran et al., 2024) builds on this idea by incorporating label text embeddings from pretrained language models as anchors, which enables the generation of more meaningful samples and enhances resistance to forgetting. While effective in mitigating forgetting, these approaches inherit significant drawbacks: training generative models is computationally expensive as image resolution increases and introduces new privacy risks (Liu et al., 2024).

Overall, existing FCL methods face practical challenges in real-world FL deployments due to privacy concerns and resource constraints. Specifically, they often: (1) assume the task identity is known during inference; (2) store exemplars from previous tasks on the server; or (3) train a generative model to synthesize replay samples. In contrast, FedProTIP requires none of these assumptions. It is explicitly designed for task-agnostic inference, where task labels are unavailable, and achieves this without replay buffers, generative models, or auxiliary task classifiers. Instead, FedProTIP leverages lightweight subspace representations for both knowledge retention and task-identity prediction, enabling effective task-agnostic FCL. This focus connects to the broader literature on class-incremental learning (CIL). (Kim et al., 2022b) provides a theoretical perspective that

decomposes the CIL problem into within-task classification and task-identity prediction, showing that both are necessary and sufficient conditions for strong performance. While centralized methods address the challenge of task-agnostic inference via out-of-distribution detection (Kim et al., 2022b;a), per-class classifiers or generative models (Zając et al., 2024; Van De Ven et al., 2021), or supervised contrastive learning with nearest-class-mean classifiers (Mai et al., 2021), none extend naturally to federated environments. Our work is the first to bring task-identity prediction into the FCL paradigm, providing a replay-free, privacy-preserving solution in the challenging task-agnostic CIL setting.

## 2.2 GRADIENT PROJECTION IN CONTINUAL LEARNING

Gradient projection methods (Zeng et al., 2019; Farajtabar et al., 2020; Chaudhry et al., 2020) for continual learning mitigate forgetting by updating model parameters in directions orthogonal to those associated with previous tasks, thereby eliminating the need to store raw data or train generative models. GPM (Saha et al., 2021) extends these approaches by extracting the bases of low-dimensional subspaces spanned by prior task representations and constraining new gradients to lie orthogonal to these subspaces. A series of follow-up works, including TRGP (Lin et al., 2022b), CUBER (Lin et al., 2022a), SGP (Saha & Roy, 2023) and DualGPM (Liang & Li, 2023a), relax the orthogonality constraints introduced in GPM to better balance stability and plasticity, which leads to improved continual learning performance. However, directly extending gradient projection methods to federated continual learning (FCL) is nontrivial due to the decentralized nature of data and limited communication budgets. FOT (Bakman et al., 2024) represents a recent attempt to adapt GPM to the FCL setting by requiring clients to share their raw feature embeddings with a central server to construct gradient subspaces. This approach introduces privacy risks and communication overhead, and its effectiveness is limited to scenarios where task identities are known at inference time – an impractical assumption in many real-world settings. In contrast, FedProTIP achieves state-of-the-art FCL performance using a communication- and computation-efficient projection scheme, without exposing raw embeddings or relying on task labels during inference.

## 3 BACKGROUND AND PROBLEM SETUP

### 3.1 PROBLEM FORMULATION

We consider the problem of continually fine-tuning a global model on streaming data $\mathcal{D}^{(t)} = \{\mathbf{x}_i^{(t)}, \mathbf{y}_i^{(t)}\}_{i=1}^{|\mathcal{D}^{(t)}|}$ distributed across $K$ client devices such that $\mathcal{D}^{(t)} = \mathcal{D}_1^{(t)} \cup \cdots \cup \mathcal{D}_K^{(t)}$. In the *domain-incremental* setting, the input distributions of two tasks, $\mathcal{X}^{(t_1)}$ and $\mathcal{X}^{(t_2)}$, are significantly different, while the label space may remain the same. In the *class-incremental* setting, the label sets of any two tasks are disjoint, i.e., $\mathcal{Y}^{(t_1)} \cap \mathcal{Y}^{(t_2)} = \emptyset$ for all $t_1 \neq t_2$. When learning a new task, data from earlier tasks is assumed to be inaccessible. The goal of federated continual learning is to obtain a global model $\mathbf{W}^{(T)}$ that minimizes the average empirical loss across $T$ tasks,

$$\min_{\mathbf{W}} \frac{1}{T} \sum_{t=1}^{T} \sum_{k=1}^{K} p_k^{(t)} \mathcal{L}_t(\mathbf{W}, \mathcal{D}_k^{(t)}), \tag{1}$$

where $p_k^{(t)}$ denotes the weight assigned to client $k$ on task $t$, and $\mathcal{L}_t$ is the empirical loss for task $t$ on local data. During inference, **task identities are not revealed** to the model.

### 3.2 GRADIENT PROJECTION MEMORY

Gradient projection memory (Saha et al., 2021) is a replay-free CL scheme that requires storing only a set of core bases $\boldsymbol{\Phi}_l^{(1:t)}$ extracted from layer-wise activations after fine-tuning the model on $t$ tasks. Specifically, let $\mathbf{W}_l^{(t)} \subset \mathbf{W}^{(t)}$ denote the parameters of layer $l$ after training on task $t$, and let $\mathbf{a}_l^{(t)} \in \mathbb{R}^{d_l \times m}$ represent the input activations to layer $l$ for $m$ training samples $\mathbf{x}^{(t)}$, where $d_l$ is the dimensionality of the activations. By applying singular value decomposition (SVD), GPM extracts a set of orthonormal bases $\boldsymbol{\Phi}_l^{(t)} \in \mathbb{R}^{d_l \times r_l^{(t)}}$ that span the dominant subspace of task $t$ activations and aggregates them with the existing bases $\boldsymbol{\Phi}_l^{(1:t-1)}$. During training on the $(t+1)$-th task, the parameter update for layer $l$, denoted $\Delta \mathbf{W}_l^{(t+1)}$, is projected onto the orthogonal complement of the subspace spanned by $\boldsymbol{\Phi}_l^{(1:t)}$,

$$\Delta \tilde{\mathbf{W}}_l^{(t+1)} \leftarrow \mathbf{Proj}_{\perp \boldsymbol{\Phi}_l^{(1:t)}} \left( \Delta \mathbf{W}_l^{(t+1)} \right). \tag{2}$$

Figure 1: **Overview of FedProTIP.** (1) Clients apply projected gradient descent; the server aggregates updates. (2) Clients extract core bases via SVD; the server merges them into a global subspace. (3) At inference, task identity is predicted by comparing test relevance vectors to stored task references.

Let $\mathbf{h}_l^{(\tau)} = \boldsymbol{\sigma}_l(\mathbf{W}_l^{(T)} \cdot \mathbf{a}_l^{(\tau)})$ denote the output activations for task $\tau$ ($\tau < T$) after training on $T$ tasks, where $\boldsymbol{\sigma}_l(\cdot)$ is the activation function at layer $l$. It follows from Eq. 2 that

$$\mathbf{h}_l^{(\tau)} = \boldsymbol{\sigma}_l \left( \mathbf{W}_l^{(\tau)} \cdot \mathbf{a}_l^{(\tau)} + \sum_{t=\tau+1}^{T} \Delta \tilde{\mathbf{W}}_l^{(t)} \cdot \mathbf{a}_l^{(\tau)} \right) \approx \boldsymbol{\sigma}_l \left( \mathbf{W}_l^{(\tau)} \cdot \mathbf{a}_l^{(\tau)} \right), \tag{3}$$

implying that subsequent updates do not significantly alter the representations learned on task $\tau$.

## 4 METHODOLOGY

While GPM has proven effective in centralized continual learning, extending it to federated settings poses major challenges. FOT (Bakman et al., 2024) offers an early adaptation by having clients share layer-wise intermediate activations, which the server uses to extract core bases. However, this approach raises significant privacy concerns, as such activations can be exploited in gradient inversion attacks (Geiping et al., 2020; Chen & Vikalo, 2024). It also introduces substantial communication overhead due to the high dimensionality of the transmitted activations.

FOT performs standard local training on client devices and applies orthogonal projections to the global model update $\Delta \mathbf{W}^{(t)} = \sum_{k=1}^{K} p_k^{(t)} \Delta \mathbf{W}_k^{(t)}$ to mitigate feature interference across tasks. However, since local models are not trained with orthogonal constraints, this mismatch can degrade performance. Moreover, like most GPM-based methods, FOT assumes task identities are known during inference, which is unrealistic in many real-world deployments. In contrast, FedProTIP avoids both task ID reliance and the collection of intermediate activations, yet delivers strong performance under task-agnostic inference.

### 4.1 LOCAL TRAINING WITH GRADIENT PROJECTION

As previously discussed, projecting the aggregated global update onto an orthogonal subspace may lead to information loss due to misalignment with client-specific gradients. Instead, FedProTIP applies projected gradient descent locally on each client and training batch according to

$$\nabla \tilde{\mathbf{W}}_k^{(t)} = \nabla \mathbf{W}_k^{(t)} - \boldsymbol{\Phi}^{(1:t-1)} \left( \boldsymbol{\Phi}^{(1:t-1)} \right)^{\top} \nabla \mathbf{W}_k^{(t)}, \tag{4}$$

$$\mathbf{W}_k^{(t)} = \mathbf{W}_k^{(t)} - \eta \nabla \tilde{\mathbf{W}}_k^{(t)}, \tag{5}$$

where $\nabla \mathbf{W}_k^{(t)}$ denotes the gradient computed from client $k$'s local data, and $\boldsymbol{\Phi}^{(1:t-1)}(\boldsymbol{\Phi}^{(1:t-1)})^{\top}$ represents the projection matrix onto the subspace spanned by core bases from earlier tasks. (The layer index $l$ has been omitted from subscripts for the sake of simplicity.) The operation in Eq. 4 can be interpreted as removing gradient components aligned with past task subspaces, thereby reducing interference with prior knowledge. Since the global projection operator $P = I - \Phi\Phi^{\top}$ is an orthogonal contraction, it never amplifies heterogeneity across clients but only preserves or reduces variance in their updates. Convergence analysis of this local training is provided in Appendix B.

### 4.2 EXTRACTING LOCAL CORE BASES

After completing projected gradient descent-based local training, clients follow the standard federated learning protocol by sending local model updates to the server for aggregation. The server then

---

**Algorithm 1:** FedProTIP Training Procedure

---

**Input:** $K$ clients, $T$ tasks, global round $E$, local datasets $\mathcal{D}_k^{(t)}$, $k \in [K]$.

**Output:** The global model $\mathbf{W}^{(T)}$, stored representations $\mathbf{\Phi}^{(1:T)}$, references $\boldsymbol{\omega}_k^{(t)}, \forall k \in [K], t \in [T]$.

1 **Initialization**: Broadcast $\mathbf{W}^{(0)}$ to all clients.;

2 **for** $t = 1, \ldots, T$ **do**

3      $\mathbf{W}^{(t,0)} \leftarrow \mathbf{W}^{(t-1)}, \mathbf{\Phi}^{(0)} \leftarrow \emptyset$;

4      **for** $e = 1, \ldots, E$ **do**

5          **for** $k \in [K]$ **do**

6              $\mathbf{W}_k^{(t,e)} \leftarrow \mathbf{PGD}(\mathbf{W}^{(t,e-1)}, \mathcal{D}_k^{(t)}, \mathbf{\Phi}^{(0:t-1)})$; /* Following Eqs. (4)-(5) */

7              Send $\mathbf{W}_k^{(t,e)}$ to the server;

8          **end**

9          $\mathbf{W}^{(t,e)} \leftarrow \sum_{k=1}^{K} p_k^{(t)} \mathbf{W}_k^{(t,e)}$;

10      **end**

11      **for** $k \in [K]$ **do**

12          $\mathbf{U}_k^{(t)}, \mathbf{a}_{L,k}^{(t)} \leftarrow \mathbf{ExtractBases}(\mathbf{W}^{(t,E)}, \mathcal{D}_k^{(t)}, \boldsymbol{\epsilon})$; /* Extract bases (Sec 4.2) */

13          $\boldsymbol{\omega}_k^{(1:t)} \leftarrow \mathbf{UpdateReference}(\mathbf{U}_k^{(t)}, \mathbf{a}_{L,k}^{(t)}, \boldsymbol{\omega}_k^{(1:t-1)})$;

14          Send $\mathbf{U}_k^{(t)}, \boldsymbol{\omega}_k^{(1:t)}$ to the server;

15      **end**

16      $\mathbf{\Phi}^{(t)} \leftarrow \mathbf{GlobalAggregate}(\mathbf{U}_1^{(t)}, \ldots, \mathbf{U}_K^{(t)})$; $\mathbf{W}^{(t)} \leftarrow \mathbf{W}^{(t,E)}$;

17 **end**

---

broadcasts the updated global model $\mathbf{W}^{(t)}$ to all clients, which proceed to perform local core basis extraction. Following the GPM strategy (Saha et al., 2021), each client $k$ samples $m$ examples from its local dataset $\mathcal{D}_k^{(t)}$, feeds them through the received model $\mathbf{W}^{(t)}$, and collects layer-wise intermediate activations. To reduce storage and communication overhead, we introduce a random activation sampling step: from the $m$ activations, a smaller subset of size $m^s \ll m$ is randomly selected, yielding $\mathbf{a}_l^{(t)} \in \mathbb{R}^{d_l \times m^s}$. These activations are projected onto the orthogonal complement of the previously learned feature subspace by subtracting their component along the existing bases,

$$\tilde{\mathbf{a}}_l^{(t)} = \mathbf{a}_l^{(t)} - \mathbf{\Phi}^{(1:t-1)} \left( \mathbf{\Phi}^{(1:t-1)} \right)^\top \mathbf{a}_l^{(t)}. \tag{6}$$

The projected activations $\tilde{\mathbf{a}}_l^{(t)}$ are then decomposed using singular value decomposition (SVD),

$$\tilde{\mathbf{a}}_l^{(t)} = \mathbf{U}_l^{(t)} \mathbf{\Sigma}_l^{(t)} \left( \mathbf{V}_l^{(t)} \right)^\top, \tag{7}$$

where $\mathbf{U}_l^{(t)} \in \mathbb{R}^{d_l \times d_l}$ is a unitary matrix and $\mathbf{\Sigma}_l^{(t)} \in \mathbb{R}^{d_l \times m}$ is a diagonal matrix of singular values. To extract the top-$r_l$ core bases from $\mathbf{U}_l^{(t)}$, a layer-specific threshold $\epsilon_l$ is applied to select the smallest number of leading components such that

$$\mathbf{U}^{(t)} = \{\mathbf{U}_l^{(t)}[\cdot, 1 : r_l] \text{ s.t.} \sum_{i=1}^{r_l} \sigma_{l,i} \geq \epsilon_l, \text{ for } \forall l \leq L\}, \tag{8}$$

with $L$ denoting the number of layers and $\sigma_{l,i}$ the $i$-th diagonal element of $\mathbf{\Sigma}_l^{(t)}$. Finally, each client $k$ sends its extracted local core bases $\mathbf{U}_k^{(t)}$ to the server for aggregation.

### 4.3 UPDATING THE GLOBAL FEATURE SUBSPACE

The server collects core bases $\mathbf{U}_k^{(t)}$ from participating clients and integrates them into the global feature subspace by removing redundant components. Aggregation is initialized by setting $\mathbf{\Phi}_l^{(t)} = \mathbf{U}_{l,1}^{(t)}$ for each layer $l$, using the core bases received from the first client. The server then iteratively updates $\mathbf{\Phi}_l^{(t)}$ by orthogonalizing and appending additional bases from the remaining clients,

$$\mathbf{\Phi}_l^{(t)} = [\mathbf{\Phi}_l^{(t)}, \mathbf{U}_{l,k}^{(t)} - \mathbf{\Phi}_l^{(t)} \left( \mathbf{\Phi}_l^{(t)} \right)^\top \mathbf{U}_{l,k}^{(t)}], \ k = 2, \ldots, K, \tag{9}$$

ensuring that the added bases are orthogonal to the current global subspace. Following aggregation, the updated global bases $\mathbf{\Phi}^{(t)}$ are broadcast to clients for the next training round, as seen in Section 4.1.

## 4.4 TASK IDENTITY INFERENCE VIA SUBSPACE RELEVANCE

In continual learning, the feature extractor (encoder) is fine-tuned across sequential tasks, while the decision layer $f_{1:t}(\cdot)$ expands as new tasks are introduced. For example, in class-incremental settings, the output dimensionality of the softmax layer grows with the number of classes. Prior works (Saha et al., 2021; Bakman et al., 2024) assume that the task identity $\tau$ is known at inference time, so predictions can be routed through the corresponding decision head $f_\tau(\cdot)$. However, this assumption is unrealistic in real-world deployments, where task labels are typically unavailable (Kim et al., 2022b).

To address this, FedProTIP introduces a task inference mechanism built on two key concepts: task relevance and task reference. Relevance quantifies how well a test input aligns with the feature subspace of each learned task, while reference vectors capture the expected relevance patterns for known tasks. As shown in Figure 1, each client constructs these reference vectors from training data by measuring how its final-layer activations relate to past task subspaces. At inference, the model computes a relevance vector for the test input and compares it to stored references, inferring task identity based on the highest aggregated similarity across clients.

**Client-side reference vector computation.** During local training, each client collects layer-wise activations $\mathbf{a}_l^{(\tau)}$ and extracts core bases $\mathbf{U}_l^{(\tau)}$ for each task $\tau$. For task-identity prediction, we use $\mathbf{a}_L^{(\tau)}$, the input activation to the final layer. After completing $T$ tasks, each client computes a reference vector $\boldsymbol{\omega}^{(\tau)} = [\omega^{(\tau,1)}, \ldots, \omega^{(\tau,T)}]$ for every $\tau \leq T$, where

$$\omega^{(\tau,t)} = \left\| \mathbf{U}_L^{(t)} \left( \mathbf{U}_L^{(t)} \right)^\top \mathbf{a}_L^{(\tau)} \right\|_2, \quad \forall \tau \leq T, \ t \leq T. \tag{10}$$

Here, $\omega^{(\tau,t)}$ measures how strongly task $\tau$'s activation aligns with the subspace of task $t$. As noted in Section 4.2, this value is typically small when $\tau < t$ since later bases are constructed after removing earlier representations. Each client $k$ stores the full set of reference vectors $\boldsymbol{\omega}_k^{(1)}, \ldots, \boldsymbol{\omega}_k^{(T)}$, which are transmitted to the server for use during inference.

**Inference-time task prediction.** Given a test sample, the global model computes the final-layer activation $\mathbf{a}_L^{\text{te}}$ and forms a task relevance vector $\hat{\boldsymbol{\omega}} = [\hat{\omega}_1, \ldots, \hat{\omega}_T]$, where $\hat{\omega}_t = \|\mathbf{U}_L^{(t)}(\mathbf{U}_L^{(t)})^\top \mathbf{a}_L^{\text{te}}\|_2, \forall t \leq T$. The server compares this relevance vector to each client's stored reference vectors using cosine similarity,

$$\mathcal{S}_k^{(t)} = \frac{\hat{\boldsymbol{\omega}} \cdot \boldsymbol{\omega}_k^{(t)}}{\|\hat{\boldsymbol{\omega}}\| \cdot \|\boldsymbol{\omega}_k^{(t)}\|}, \quad \forall k \in [K], \ t \leq T. \tag{11}$$

Each client casts a vote for the task with highest similarity, $\mathcal{V}_k = \text{argmax}_t \mathcal{S}_k^{(t)}$, and the final task identity is selected by majority vote across all clients. Despite involving multiple votes, the relevance vectors $\hat{\boldsymbol{\omega}} \in \mathbb{R}^T$ are low-dimensional, leading to negligible inference overhead. In large-scale FL systems, task prediction can be efficiently approximated using only a representative subset of clients.

## 5 EXPERIMENTS

We evaluate FedProTIP on three standard continual learning benchmarks: CIFAR100 and ImageNet-R (Hendrycks et al., 2021) for class-incremental learning, and DomainNet (Peng et al., 2019) for domain-incremental learning. Comparisons are made against six baselines: FedAvg (McMahan et al., 2017), GLFC (Dong et al., 2022), LGA (Dong et al., 2023), TARGET (Zhang et al., 2023), FOT (Bakman et al., 2024), and LANDER (Tran et al., 2024). Following (Yurochkin et al., 2019), we simulate non-IID client distributions by sampling data partitions via a Dirichlet distribution with varying concentration parameter $\alpha$ (lower $\alpha$ implies greater heterogeneity). All methods use a ResNet-18 backbone pretrained on ImageNet-1K (He et al., 2016) and fine-tuned on each benchmark dataset. Additional results using ResNets and ViTs trained from scratch are provided in Appendix A.1.

Following prior work (Chaudhry et al., 2018), we evaluate performance using two standard metrics: average accuracy (ACC) and forgetting (FT), defined as

$$\text{ACC} = \frac{1}{T} \sum_{t=1}^{T} \text{acc}_t^{(T)}, \quad \text{FT} = \frac{1}{T} \sum_{t=1}^{T-1} \left( \max_{i \in \{1, \ldots, T-1\}} \text{acc}_t^{(i)} - \text{acc}_t^{(T)} \right), \tag{12}$$

where $\text{acc}_t^{(t)}$ is the accuracy on task $t$ after training on $t$ tasks, and $\text{acc}_t^{(T)}$ is the final accuracy on task $t$ after all $T$ tasks. Task identities are not revealed during inference, consistent with the task-agnostic setting studied in this work. Additional experimental details are provided in Appendix C.

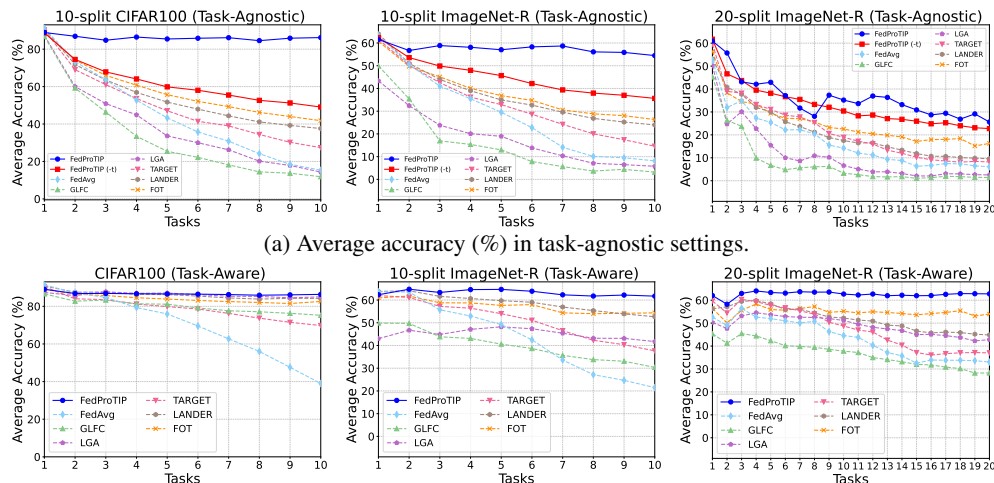

(a) Average accuracy (%) in task-agnostic settings.

(b) Average accuracy (%) in task-aware settings where true task-ID is provided during inference.

Figure 2: Average accuracy of class-incremental learning on three benchmarks. (a) Task-agnostic inference, where task identity is unknown. (b) Task-aware inference, where the true task ID is provided at test time.

## 5.1 PERFORMANCE IN TASK-AGNOSTIC AND TASK-AWARE SETTINGS

Figure 2 reports the average accuracy over all tasks seen so far ($y$-axis) as a function of the number of learned tasks ($x$-axis) in class-incremental learning experiments on 10-split CIFAR100 and 10/20-split ImageNet-R. Notably, in the task-agnostic setting, where the true task identity of test samples is unknown (Figure 2a), FedProTIP consistently outperforms all baselines throughout the training.

While other baselines achieve competitive performance on CIFAR100 and ImageNet-R in task-aware settings (Figure 2b), they suffer severe degradation when task identities are unavailable at test time. This stems from two key issues: (1) fine-tuning on new tasks collapses previously learned feature representations, and (2) this collapse causes misalignment with task-specific decision layers. FedProTIP maintains strong performance on both recent and earlier tasks by combining two core mechanisms: (i) orthogonal gradient projection to reduce cross-task interference, and (ii) task identification to dynamically route test inputs to the appropriate output head. Notably, even without task-ID prediction, our method (labeled as FedProTIP (-t)) surpasses FOT, highlighting the effectiveness of local projection and the global subspace aggregated from core bases. While TIP is less effective on the more challenging 20-split ImageNet-R than in the 10-split setting, since each task contains fewer classes and less data to define distinctive subspaces, both variants of FedProTIP still outperform all baselines and demonstrate strong scalability as the number of tasks increases.

## 5.2 ROBUSTNESS UNDER DATA HETEROGENEITY AND FORGETTING

**Data heterogeneity.** As shown in Table 1, FedProTIP consistently outperforms all baselines across various values of the Dirichlet concentration parameter $\alpha$, which controls the degree of data heterogeneity (larger $\alpha$ corresponds to more IID-like partitions). As heterogeneity increases (i.e., smaller $\alpha$), *client drift* (Karimireddy et al., 2020) exacerbates *catastrophic forgetting*, degrading the performance of most methods. Despite this, FedProTIP remains robust. For example, on CIFAR100, while competing baselines like FOT and LANDER experience accuracy drops of 12% and 15%, respectively, when moving from IID to $\alpha = 0.2$, FedProTIP sees only a 6% decline. Moreover, it sustains high accuracy while keeping forgetting near zero, demonstrating strong resilience to heterogeneous client updates. Beyond *class-incremental* learning, we evaluate robustness under overlapping task semantics using DomainNet in a *domain-incremental* setting, where all tasks share the same label space. In the task-agnostic scenario, both the baselines and FedProTIP (-t) use a single shared classifier. Under this setup, FedProTIP (-t) consistently outperforms all competing methods across varying levels of heterogeneity, demonstrating benefits that extend beyond task identity prediction by enabling more robust representation learning. In contrast, FedProTIP maintains separate classifiers for each task and uses task prediction to route test inputs. Although this approach yields slightly lower accuracy than FedProTIP (-t), which uses a shared classifier, FedProTIP remains competitive – it achieves the lowest forgetting across all settings and delivers accuracy second only to our FedProTIP (-t).

Table 1: Accuracy (↑) and forgetting (↓) metrics (%) on 10-split CIFAR-100 and 6-split DomainNet across different heterogeneity levels (Dirichlet $\alpha$). **Bold** and underline indicate the best and second-best results, respectively. GLFC and LGA are incompatible with domain-incremental learning and are marked with $\star$. Full tables with standard deviations are provided in the appendix.

| Method | 10-Split CIFAR100 (Class-IL) | | | | | | 6-Split DomainNet (Domain-IL) | | | | | |
| | IID | | $\alpha = 0.5$ | | $\alpha = 0.2$ | | IID | | $\alpha = 0.5$ | | $\alpha = 0.2$ | |
| | ACC | FT | ACC | FT | ACC | FT | ACC | FT | ACC | FT | ACC | FT |
|---|---|---|---|---|---|---|---|---|---|---|---|---|
| FedAvg | 18.92 | 63.20 | 15.35 | 62.90 | 15.76 | 52.80 | 10.79 | 27.74 | 10.72 | 25.66 | 10.53 | 25.57 |
| GLFC | 14.07 | 69.17 | 11.86 | 68.20 | 10.33 | 63.98 | $\star$ | $\star$ | $\star$ | $\star$ | $\star$ | $\star$ |
| LGA | 14.93 | 72.06 | 14.35 | 71.09 | 11.67 | 65.82 | $\star$ | $\star$ | $\star$ | $\star$ | $\star$ | $\star$ |
| TARGET | 29.56 | 42.73 | 27.37 | 37.60 | 23.05 | 34.63 | 21.53 | 9.73 | 20.61 | 7.89 | 20.64 | 8.31 |
| LANDER | 39.09 | 9.27 | 37.59 | 10.21 | 23.56 | 13.28 | 21.88 | 8.90 | 21.59 | 10.27 | 22.11 | 8.59 |
| FOT | 46.86 | 21.11 | 41.80 | 20.86 | 34.65 | 18.09 | 24.59 | 8.85 | 24.13 | 8.44 | 23.84 | 8.33 |
| FedProTIP (-t) | 52.30 | 15.66 | 48.41 | 15.59 | 42.19 | 14.91 | **29.64** | 6.38 | **28.85** | 6.43 | **28.74** | 6.14 |
| **FedProTIP** | **87.94** | **1.30** | **86.00** | **0.83** | **81.94** | **1.35** | 27.60 | **2.89** | 25.30 | 3.76 | 25.98 | **2.88** |

Table 2: Accuracy (↑) and forgetting (↓) metrics (%) computed in the experiments on 5-split, 10-split, and 20-split ImageNet-R. **Bold** and underline indicate the best and the second-best methods, respectively.

| Method | 5-Split ImageNet-R | | | | 10-Split ImageNet-R | | | | 20-Split ImageNet-R | | | |
| | IID | | $\alpha = 0.5$ | | IID | | $\alpha = 0.5$ | | IID | | $\alpha = 0.5$ | |
| | ACC | FT | ACC | FT | ACC | FT | ACC | FT | ACC | FT | ACC | FT |
|---|---|---|---|---|---|---|---|---|---|---|---|---|
| FedAvg | 22.70 | 37.11 | 22.22 | 36.26 | 8.74 | 43.84 | 8.15 | 41.14 | 9.77 | 43.18 | 6.08 | 31.75 |
| GLFC | 7.26 | 16.99 | 7.47 | 17.12 | 3.34 | 29.88 | 3.18 | 29.80 | 2.12 | 36.12 | 1.43 | 30.40 |
| LGA | 8.33 | 21.13 | 7.38 | 19.91 | 5.84 | 36.41 | 5.76 | 35.05 | 3.32 | 43.29 | 2.52 | 40.76 |
| TARGET | 40.95 | 14.43 | 37.71 | 14.89 | 17.64 | 25.83 | 14.60 | 23.52 | 9.77 | 29.87 | 8.18 | 24.63 |
| LANDER | 35.50 | **1.45** | 36.83 | **1.46** | 24.53 | 5.39 | 23.96 | 3.10 | 12.23 | **10.33** | 8.73 | **8.00** |
| FOT | 39.77 | 13.43 | 38.58 | 13.24 | 23.68 | 14.61 | 26.31 | 15.52 | 22.50 | 16.08 | 16.27 | 13.26 |
| FedProTIP (-t) | 50.00 | 6.26 | 46.99 | 8.03 | 41.35 | 8.80 | 35.64 | 8.65 | 31.43 | 10.37 | 22.75 | 10.97 |
| **FedProTIP** | **55.65** | 3.36 | **54.49** | 6.03 | **52.68** | 10.34 | **54.48** | 7.48 | **34.80** | 12.03 | **25.62** | 12.21 |

**Catastrophic forgetting.** As shown in Table 2, FedProTIP consistently performs well on ImageNet-R across 5-, 10-, and 20-task splits. ImageNet-R ranks among the most challenging continual learning benchmarks, with many existing methods degrading to single-digit accuracy. The difficulty grows with the number of tasks, as catastrophic forgetting accumulates; this is reflected in the higher forgetting values observed in larger splits. While LANDER often achieves the lowest forgetting, it does so at the expense of significantly lower accuracy. In contrast, FedProTIP strikes a strong balance, outperforming the second-best method (FOT) by 8%–28% in accuracy while maintaining forgetting that is competitive across settings. Even in the most challenging 20-task scenario, it sustains 25%–35% accuracy – well above the baselines. These results demonstrate that FedProTIP scales more steadily with the number of tasks, a crucial property for realistic continual learning.

## 5.3 Ablation Studies

**Varying number of clients.** To evaluate how FedProTIP scales with the federated systems size, we conduct experiments with 5, 10, and 20 clients. To ensure a consistent number of participants per communication round, we set the client sampling rates to 1, 0.5, and 0.25, respectively. As shown in Table 3, FedProTIP consistently outperforms competing methods across all configurations. Increasing the number of clients typically leads to a decline in performance of all FCL methods due to greater data heterogeneity and reduced local diversity. However, FedProTIP remains robust in these settings. Its use of local gradient projection reduces interference between tasks during client updates, which helps mitigate forgetting and preserve accuracy.

**Effect of the projection threshold.** FedProTIP extracts core bases from feature representations using a layer-wise threshold $\epsilon_l$, which controls how many directions are retained per layer. We assess sensitivity to this parameter by varying $\epsilon_l \in [0.7, 0.9]$ across all layers on CIFAR100 (see Fig. 3). Results show that FedProTIP is largely insensitive to the threshold choice, maintaining stable accuracy across this range. However, very high thresholds preserve more directions from prior tasks, favoring stability but reducing plasticity, as fewer orthogonal directions remain for new tasks. This highlights a trade-off: moderate thresholds offer a better balance between preserving past knowledge and adapting to new tasks. Additional experiments on DomainNet and ImageNet-R (Appendix A.3) support these findings, showing consistent robustness alongside this stability–plasticity effect.

**Impact of task prediction strategies** We investigate whether task-agnostic continual learning methods originally developed for centralized settings can be adapted to federated scenarios. Specifi-

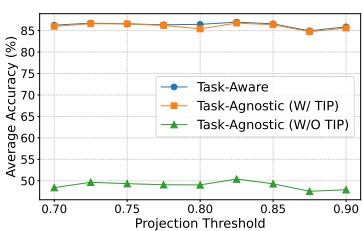

Figure 3: Effect of projection threshold $\epsilon_l$ on FedProTIP accuracy.

Table 3: Table 3: Accuracy ($\uparrow$) and forgetting ($\downarrow$) on 10-split CIFAR-100 under different numbers of clients $(5, 10, 20)$.

| Method | 10-Split CIFAR100 ($\alpha = 0.5$) | | | | | |
| | 5 | | 10 | | 20 | |
| | ACC | FT | ACC | FT | ACC | FT |
|---|---|---|---|---|---|---|
| GLFC | 11.86 | 68.20 | 9.55 | 61.04 | 7.45 | 51.42 |
| LGA | 14.35 | 71.09 | 12.24 | 69.06 | 13.36 | 63.58 |
| TARGET | 27.37 | 37.60 | 23.28 | 40.65 | 22.41 | 43.27 |
| LANDER | 37.59 | 10.21 | 26.60 | 2.16 | 23.42 | 6.13 |
| FOT | 41.80 | 20.86 | 39.73 | 13.08 | 37.35 | 13.66 |
| FedProTIP (-t) | 48.41 | 15.59 | 41.33 | 10.70 | 40.06 | 9.55 |
| **FedProTIP** | **86.00** | **0.83** | **81.34** | **0.59** | **81.10** | **0.28** |

Table 4: Comparison with federated variants of task-agnostic inference methods. $\Delta$ values denote performance gains when combined with FCL methods.

| Method | Task-Agnostic | | Task-Aware | |
| | $ACC_\Delta$ | $FT_\Delta$ | ACC | FT |
|---|---|---|---|---|
| Fed+PEC | 19.56 | 12.18 | 50.04 | 0.00 |
| Fed+SCR | 34.26 | 38.22 | – | – |
| Tar+LODE | $29.87_{\uparrow 2.50}$ | $44.09_{\uparrow 6.48}$ | 69.24 | 15.96 |
| FedProTIP | $86.00_{\uparrow 37.19}$ | $0.83_{\downarrow 14.75}$ | 86.26 | 0.96 |

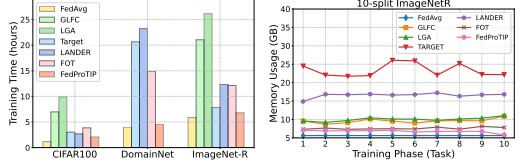

Figure 4: Training efficiency comparison: (left) training time (hours), (right) average GPU memory usage.

cally, we consider the replay-free PEC (Zając et al., 2024), which assigns a separate classifier to each class, and the replay-based SCR (Mai et al., 2021), which uses a nearest-class-mean classifier. For SCR, we average class prototypes across clients at inference time while keeping replay data local, and use FedAvg for model aggregation. As shown in Table 5, both methods yield significantly lower accuracy under task-agnostic inference. While SCR outperforms PEC, it relies on clients retaining local data, which may violate privacy constraints. We also evaluate LODE (Liang & Li, 2023b), which decouples intra- and inter-task losses to implicitly support task-agnostic inference. Applied to the generative replay method TARGET, LODE offers only marginal gains ($+2.50\%$), suggesting that naive loss decoupling is insufficient for robust performance in federated settings.

## 5.4 TRAINING TIMES, MEMORY USAGE, AND COMMUNICATION COST

We evaluate the efficiency of FCL methods along three axes: training time, GPU memory usage, and communication overhead (see Figure 4). **Training time:** FedProTIP achieves the fastest training among continual learning methods across all datasets, second only to FedAvg, which performs no CL. On high-resolution datasets like DomainNet, generative methods (e.g., GLFC, LGA, TARGET, LANDER) incur significant overhead from image reconstruction and replay. For example, FedProTIP trains up to $5\times$ faster than TARGET and LANDER. Its efficiency stems from using randomized SVD on sampled activation matrices $A_l^{(t)} \in \mathbb{R}^{d_l \times m^s}$, where $m^s \ll n$, reducing decomposition cost to $\mathcal{O}(d_l \cdot r_l^2)$ and requiring only $\mathcal{O}(d_l \cdot r_l)$ operations per layer for gradient projection ($r_l$ denotes the number of retained singular vectors). **Memory usage:** Clients store only the retained core bases $U_l^{(t)}[1 : r_l] \in \mathbb{R}^{d_l \times r_l}$ for each task and layer, resulting in memory usage of $\mathcal{O}(d_l \cdot r_l)$. Extended results are shown in Figure 5. **Communication cost:** After local training, clients send only core bases to the server, incurring per-layer cost $\mathcal{O}(d_l \cdot r_l)$; this is significantly lower than the $\mathcal{O}(d_l \cdot s_l)$ required by FOT (Bakman et al., 2024), where e.g. in CIFAR100 $s_l = 5d_l$. The actual per-task cost comparison is reported in Appendix A.5. Notably, since the number of retained bases $r_l$ tends to decrease over time, both memory and communication overheads diminish throughout training, making FedProTIP highly scalable.

## 6 CONCLUSION

We proposed FedProTIP, a federated continual learning (FCL) framework that leverages gradient projection to reduce feature interference and mitigate catastrophic forgetting. Unlike prior FCL methods, FedProTIP requires neither storing past data nor training generative models for rehearsal. FedProTIP extracts core feature subspaces via memory-efficient randomized SVD and uses them to predict task identity, enabling better alignment between test inputs and decision layers. Extensive experiments across three benchmark datasets show that FedProTIP consistently outperforms state-of-the-art methods while maintaining lower computational overhead.

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

# A ADDITIONAL EXPERIMENTAL RESULTS

## A.1 RESULTS ON DIFFERENT MODELS

We report the results using a scratch-trained ResNet18 in Table 5. In the task-agnostic inference setting, our method achieves the best performance, showing a significant margin over all other baselines. We set the task identity prediction threshold to $\epsilon_l = 0.95$, $\forall l$, based on a hyperparameter search. This threshold is higher than the one used for the pretrained ResNet18 model ($\epsilon_l = 0.7$), as the pretrained model provides a stronger feature extractor that better generalizes across tasks. In contrast, when training from scratch, preserving knowledge of previous tasks becomes more critical, hence the need for a higher threshold.

As shown in Table 5, while FedProTIP is not the best-performing method in the task-aware inference scenario, where the ground-truth task ID is available during testing, it outperforms all baselines in the more practical task-agnostic setting with a large margin. Our method achieves the highest accuracy and lowest forgetting, primarily due to effective task identity prediction.

We also evaluate our method on a different backbone, pre-trained ViT-B/16 (Steiner et al., 2021), as reported in Table 6. In this setting, CIFAR100 images ($3 \times 32 \times 32$) are resized to $224 \times 224$ to match the ViT input resolution. We set the number of local epochs to 5 and perform 20 global rounds per task. For TARGET and LANDER, we observed that generating synthetic images at the native $32 \times 32$ resolution and subsequently applying resizing augmentation yields better performance, and we adopt this strategy in our experiments.

Table 5: Metrics (%) of accuracy ($\uparrow$) and forgetting ($\downarrow$) computed in the experiments on 10-split CIFAR100 ($\alpha = 0.5$) using **ResNet18 from scratch**. We report the average accuracy and standard deviation over 3 trials, each with different seeds. $\epsilon_l = 0.95$ is used for FedProTIP.

| Method | Task-Agnostic | | Task-Aware | |
|---|---|---|---|---|
| | ACC | FT | ACC | FT |
| FedAvg | $11.04_{\pm 0.37}$ | $54.90_{\pm 1.62}$ | $36.87_{\pm 1.36}$ | $42.69_{\pm 1.00}$ |
| GLFC | $6.04_{\pm 0.22}$ | $46.76_{\pm 1.09}$ | $36.93_{\pm 8.68}$ | $16.81_{\pm 6.81}$ |
| LGA | $6.96_{\pm 0.43}$ | $53.31_{\pm 0.83}$ | $50.75_{\pm 3.10}$ | $10.21_{\pm 2.81}$ |
| Target | $23.05_{\pm 1.93}$ | $\underline{9.00}_{\pm 1.15}$ | $\underline{71.86}_{\pm 0.55}$ | $2.32_{\pm 0.66}$ |
| Lander | $\underline{29.37}_{\pm 1.09}$ | $20.17_{\pm 1.90}$ | $\mathbf{73.69}_{\pm 0.64}$ | $1.39_{\pm 0.38}$ |
| FOT | $22.18_{\pm 1.35}$ | $9.10_{\pm 0.57}$ | $67.07_{\pm 1.36}$ | $\underline{0.73}_{\pm 0.06}$ |
| FedProTIP (-t) | $24.84_{\pm 0.91}$ | $12.29_{\pm 1.54}$ | $68.75_{\pm 1.71}$ | $\mathbf{0.68}_{\pm 0.48}$ |
| FedProTIP | $\mathbf{65.77}_{\pm 1.92}$ | $\mathbf{2.38}_{\pm 0.75}$ | — | — |

Table 6: Metrics (%) of accuracy ($\uparrow$) and forgetting ($\downarrow$) on 10-split CIFAR100 using **ViT-B/16**. We report the average and standard deviation over 2 trials with different seeds. $\epsilon_l = 0.7$ is used for FedProTIP.

| Method | Task-Agnostic | | Task-Aware | |
|---|---|---|---|---|
| | ACC | FT | ACC | FT |
| FedAvg | $67.15_{\pm 4.40}$ | $22.34_{\pm 0.98}$ | $95.18_{\pm 0.69}$ | $2.88_{\pm 0.33}$ |
| Target | $\underline{81.50}_{\pm 0.06}$ | $7.33_{\pm 1.34}$ | $\underline{98.30}_{\pm 0.12}$ | $\underline{0.37}_{\pm 0.01}$ |
| Lander | $61.43_{\pm 5.53}$ | $27.33_{\pm 4.31}$ | $96.07_{\pm 1.05}$ | $2.39_{\pm 0.97}$ |
| FOT | $72.27_{\pm 0.79}$ | $21.73_{\pm 0.46}$ | $96.46_{\pm 0.20}$ | $2.28_{\pm 0.15}$ |
| FedProTIP (-t) | $79.90_{\pm 1.46}$ | $\underline{4.54}_{\pm 0.35}$ | $\mathbf{98.36}_{\pm 0.04}$ | $\mathbf{0.15}_{\pm 0.09}$ |
| FedProTIP | $\mathbf{98.38}_{\pm 0.02}$ | $\mathbf{0.20}_{\pm 0.08}$ | — | — |

## A.2 BATCH SIZE SENSITIVITY

We evaluate FedProTIP and baselines with batch sizes 32, 64, and 128 (Table 8) under both task-aware and task-agnostic inference. Across all settings, FedProTIP consistently outperforms prior methods. The relative gain from TIP is smaller at low batch sizes, since limited samples increase the variance of final-layer activations, injecting noise into the relevance vector and cosine similarities. As batch size grows, variance decreases, stabilizing TIP and amplifying its benefits. Even in the small-batch regime, however, FedProTIP still yields meaningful improvements under task-agnostic inference.

Table 7: Task prediction accuracy at each training phase on 10-split CIFAR100, 6-split DomainNet, and 10-split ImageNet-R at $\alpha = 0.5$.

| Dataset | T1 | T2 | T3 | T4 | T5 | T6 | T7 | T8 | T9 | T10 |
|---|---|---|---|---|---|---|---|---|---|---|
| 10-split CIFAR100 | 1 | 1 | 0.978 | 1 | 1 | 0.989 | 0.997 | 0.989 | 1 | 0.996 |
| 5-split ImageNet-R | 1 | 1 | 0.836 | 0.944 | 0.885 | – | – | – | – | – |
| 10-split ImageNet-R | 1 | 0.895 | 0.940 | 0.912 | 0.878 | 0.906 | 0.943 | 0.904 | 0.896 | 0.874 |
| 20-split ImageNet-R | 1 | 0.9445 | 0.700 | 0.685 | 0.682 | 0.600 | 0.518 | 0.456 | 0.588 | 0.556 |
| (T11-T20) | 0.541 | 0.577 | 0.572 | 0.517 | 0.484 | 0.455 | 0.458 | 0.42 | 0.456 | 0.406 |
| 6-split DomainNet (order 1) | 1 | 1 | 0.673 | 0.88 | 0.8945 | 0.8765 | - | - | - | - |
| 6-split DomainNet (order 2) | 1 | 0.996 | 0.755 | 0.934 | 0.868 | 0.923 | - | - | - | - |

Table 8: Metrics computed in the experiments on 10-Split CIFAR100 with $\alpha = 0.5$ and varying batch sizes $\{32, 64, 128\}$.

| Batch Size | Method | Task-Aware | Task-Agnostic | +TIP |
|---|---|---|---|---|
| | FedAvg | 24.82 | 10.74 | - |
| | GLFC | 81.56 | 13.28 | - |
| | LGA | 82.59 | 13.03 | - |
| 32 | TARGET | 74.29 | 31.67 | - |
| | LANDER | 78.33 | 32.30 | - |
| | FOT | 83.38 | 41.05 | - |
| | FedProTIP | **87.86** | **48.40** | 84.22 |
| | FedAvg | 38.99 | 15.35 | - |
| | GLFC | 75.26 | 22.86 | - |
| | LGA | 85.04 | 14.35 | - |
| 64 | TARGET | 69.81 | 27.37 | - |
| | LANDER | 84.19 | 37.59 | - |
| | FOT | 82.59 | 41.80 | - |
| | FedProTIP | **86.26** | **48.41** | 86.00 |
| | FedAvg | 53.65 | 20.67 | - |
| | GLFC | 73.25 | 12.40 | - |
| | LGA | 75.25 | 11.53 | - |
| 128 | TARGET | 67.58 | 26.70 | - |
| | LANDER | 79.83 | 30.30 | - |
| | FOT | 81.61 | 39.94 | - |
| | FedProTIP | **85.20** | **45.80** | 85.20 |

## A.3 DIFFERENT THRESHOLD VALUES IN FEDPROTIP

We present the results of FedProTIP with different threshold values on CIFAR100, DomainNet, and ImageNet-R in Table 9, evaluating thresholds of 0.7, 0.8, and 0.9 in terms of both average accuracy and forgetting. Across all three datasets, FedProTIP maintains stable accuracy in both task-aware and task-agnostic settings, showing only minor sensitivity to the choice of threshold.

On CIFAR100, forgetting in the task-agnostic case remains positive but steadily decreases as the threshold increases, while on ImageNet-R, a similar trend is observed, culminating in negative forgetting at $\epsilon_l = 0.9$. Negative forgetting arises because the task-identity predictor improves as more tasks are introduced, retroactively correcting earlier misclassifications. At early stages, the predictor is poorly calibrated and often misassigns samples from earlier tasks, but later tasks provide richer contrast and sharpen decision boundaries, boosting measured accuracy on prior tasks. The threshold parameter $\epsilon_l$ also plays a critical role. A higher threshold enforces stricter preservation of gradient subspaces, biasing the stability–plasticity trade-off toward stability. In practice, this means that representations associated with earlier tasks are less likely to be overwritten when new tasks arrive. As a result, catastrophic forgetting is reduced, and in some cases (e.g., ImageNet-R at $\epsilon_l = 0.9$) the combination of preserved subspaces and improved task-identity prediction even yields negative forgetting.

Table 9: Metrics computed from FedProTIP experiments on 20-Split DomainNet and 10-Split ImageNet-R ($\alpha = 0.5$) with different thresholds $\epsilon_l$.

| Dataset | Threshold | Task-Aware | | Task-Agnostic | | +TIP | |
|---|---|---|---|---|---|---|---|
| | | ACC | FT | ACC | FT | ACC | FT |
| 10-split CIFAR100 | 0.7 | 86.26 | 1.23 | 86.00 | 15.59 | 48.41 | 1.26 |
| | 0.8 | 86.46 | 0.38 | 85.40 | 12.90 | 49.04 | 1.35 |
| | 0.9 | 85.88 | 0.08 | 85.59 | 11.80 | 47.91 | 0.15 |
| 10-split ImageNet-R | 0.7 | 61.72 | 2.35 | 54.48 | 7.48 | 35.64 | 8.65 |
| | 0.8 | 61.37 | 3.07 | 59.19 | 1.38 | 37.18 | 8.07 |
| | 0.9 | 58.99 | 1.63 | 53.41 | -2.16 | 33.80 | 6.35 |
| 6-split DomainNet | 0.7 | 28.75 | 1.45 | 28.85 | 6.43 | 25.30 | 3.76 |
| | 0.8 | 29.20 | 1.06 | 29.21 | 5.36 | 27.97 | 1.53 |
| | 0.9 | 28.99 | 0.72 | 27.35 | 6.41 | 27.78 | 0.76 |

Table 10: Metrics (%) of accuracy ($\uparrow$) and forgetting ($\downarrow$) computed in the experiments on 10-split CIFAR100. We report the average accuracy and standard deviation over 3 trials, each with different seeds.

| | 10-Split CIFAR100 | | | | | |
|---|---|---|---|---|---|---|
| Method | IID | | $\alpha = 0.5$ | | $\alpha = 0.2$ | |
| | ACC | FT | ACC | FT | ACC | FT |
| FedAvg | $18.92_{\pm 2.45}$ | $63.20_{\pm 1.36}$ | $15.35_{\pm 2.82}$ | $62.90_{\pm 0.79}$ | $15.76_{\pm 7.28}$ | $52.80_{\pm 4.36}$ |
| GLFC | $14.07_{\pm 1.10}$ | $69.17_{\pm 0.31}$ | $11.86_{\pm 2.00}$ | $68.20_{\pm 2.36}$ | $10.33_{\pm 1.97}$ | $63.98_{\pm 1.96}$ |
| LGA | $14.93_{\pm 1.09}$ | $72.06_{\pm 1.44}$ | $14.35_{\pm 1.07}$ | $71.09_{\pm 2.65}$ | $11.67_{\pm 0.65}$ | $65.82_{\pm 1.20}$ |
| TARGET | $29.56_{\pm 0.75}$ | $42.73_{\pm 4.95}$ | $27.37_{\pm 1.00}$ | $37.60_{\pm 5.30}$ | $23.05_{\pm 2.56}$ | $34.63_{\pm 2.74}$ |
| LANDER | $39.09_{\pm 1.99}$ | $9.27_{\pm 1.11}$ | $37.59_{\pm 3.85}$ | $10.21_{\pm 1.59}$ | $23.56_{\pm 5.61}$ | $13.28_{\pm 4.17}$ |
| FOT | $46.86_{\pm 2.67}$ | $21.11_{\pm 0.87}$ | $41.80_{\pm 1.12}$ | $20.86_{\pm 1.12}$ | $34.65_{\pm 1.39}$ | $18.09_{\pm 0.72}$ |
| FedProTIP (-t) | $52.30_{\pm 1.81}$ | $15.66_{\pm 0.77}$ | $48.41_{\pm 0.51}$ | $15.59_{\pm 0.80}$ | $42.19_{\pm 0.97}$ | $14.91_{\pm 1.11}$ |
| **FedProTIP** | $\mathbf{87.94}_{\pm 0.79}$ | $\mathbf{0.34}_{\pm 0.59}$ | $\mathbf{86.00}_{\pm 0.75}$ | $\mathbf{0.83}_{\pm 0.47}$ | $\mathbf{81.94}_{\pm 1.02}$ | $\mathbf{1.35}_{\pm 0.47}$ |

Finally, across all datasets and thresholds, the with TIP setting shows minimal sensitivity to threshold choice in terms of accuracy. However, excessively high thresholds can overemphasize stability, limiting plasticity and thereby reducing the learnability of new tasks.

## A.4 DIFFERENT TASK ORDERS

We present results for different task orderings in DomainNet. Table 1 of the main paper, the task order is as follows: (clipart $\rightarrow$ real $\rightarrow$ painting $\rightarrow$ sketch $\rightarrow$ infograph $\rightarrow$ quickdraw). Recognizing that DomainNet exhibits varying levels of task/domain similarity, we include Table 11 to report results under a second ordering: (clipart $\rightarrow$ infograph $\rightarrow$ painting $\rightarrow$ quickdraw $\rightarrow$ real $\rightarrow$ sketch). These results show that FedProTIP consistently achieves strong performance regardless of task order, highlighting its robustness to domain heterogeneity and variations in task scheduling. This trend holds across both orderings, with FedProTIP without TIP providing greater advantages in domain-incremental learning. Adapting domain-incremental specific modules in conjunction with TIP represents a promising direction for future research.

## A.5 COMMUNICATION COST

Table 14 reports the communication cost per task and per client. For fair comparison, we set the same sampling dimension of the activation matrix for FOT and FedProTIP. The "Act" row corresponds to FOT, which uploads randomized activations at every round, incurring a fixed 48 MB overhead per task. In contrast, FedProTIP communicates only compact core bases and reference vectors. Together, these remain far smaller (<10 MB for bases and negligible for references), and the cost of bases further decreases as fewer are extracted in later tasks (Sec. 5.4). As a result, FedProTIP reduces client-to-server communication by an order of magnitude while preserving accuracy, making it far more scalable under bandwidth constraints.

Table 11: Metrics (%) of accuracy (↑) and forgetting (↓) computed in the experiments on 6-split DomainNet of order (clipart → real → painting → sketch → infograph → quickdraw). We report the average accuracy and standard deviation over 2 trials, each with different seeds.

| Method | 20-Split DomainNet | | | | | |
| | IID | | $\alpha = 0.5$ | | $\alpha = 0.2$ | |
| | ACC | FT | ACC | FT | ACC | FT |
|---|---|---|---|---|---|---|
| FedAvg | $10.79_{\pm 0.18}$ | $27.74_{\pm 0.83}$ | $10.72_{\pm 0.14}$ | $25.66_{\pm 0.47}$ | $10.53_{\pm 0.20}$ | $25.57_{\pm 0.27}$ |
| TARGET | $21.53_{\pm 0.93}$ | $9.73_{\pm 0.51}$ | $20.61_{\pm 0.18}$ | $7.89_{\pm 0.54}$ | $20.64_{\pm 0.90}$ | $8.31_{\pm 1.12}$ |
| LANDER | $21.88_{\pm 0.32}$ | $8.90_{\pm 0.13}$ | $21.59_{\pm 1.00}$ | $10.27_{\pm 0.75}$ | $22.11_{\pm 0.26}$ | $8.59_{\pm 0.85}$ |
| FOT | $24.59_{\pm 1.00}$ | $8.85_{\pm 0.30}$ | $24.13_{\pm 0.25}$ | $8.44_{\pm 0.31}$ | $23.84_{\pm 0.00}$ | $8.33_{\pm 0.15}$ |
| FedProTIP (-t) | $\mathbf{29.64}_{\pm 0.86}$ | $\underline{6.38}_{\pm 0.09}$ | $\mathbf{28.85}_{\pm 1.46}$ | $\underline{6.43}_{\pm 0.54}$ | $\mathbf{28.74}_{\pm 0.17}$ | $\underline{6.14}_{\pm 0.33}$ |
| **FedProTIP** | $\underline{27.60}_{\pm 0.91}$ | $\mathbf{2.89}_{\pm 0.43}$ | $\underline{25.30}_{\pm 0.30}$ | $\mathbf{3.76}_{\pm 1.14}$ | $\underline{25.98}_{\pm 0.18}$ | $\mathbf{2.88}_{\pm 0.01}$ |

Table 12: Metrics (%) of accuracy (↑) and forgetting (↓) computed in the experiments on 6-split DomainNet of order (clipart → infograph → painting → quickdraw → real → sketch). We report the average accuracy and standard deviation over 2 trials, each with different seeds.

| Method | 20-Split DomainNet | | | | | |
| | IID | | $\alpha = 0.5$ | | $\alpha = 0.2$ | |
| | ACC | FT | ACC | FT | ACC | FT |
|---|---|---|---|---|---|---|
| FedAvg | $20.34_{\pm 0.74}$ | $17.16_{\pm 0.33}$ | $20.53_{\pm 0.42}$ | $15.91_{\pm 0.30}$ | $20.11_{\pm 0.25}$ | $15.40_{\pm 0.85}$ |
| TARGET | $26.53_{\pm 0.23}$ | $3.62_{\pm 0.51}$ | $25.97_{\pm 0.57}$ | $3.08_{\pm 0.27}$ | $25.65_{\pm 1.10}$ | $3.08_{\pm 0.76}$ |
| LANDER | $26.06_{\pm 0.19}$ | $\underline{2.32}_{\pm 0.04}$ | $25.45_{\pm 0.15}$ | $\underline{2.70}_{\pm 0.16}$ | $25.31_{\pm 0.10}$ | $\underline{2.21}_{\pm 0.31}$ |
| FOT | $28.57_{\pm 0.11}$ | $6.31_{\pm 0.11}$ | $28.79_{\pm 0.76}$ | $6.01_{\pm 0.39}$ | $28.33_{\pm 0.56}$ | $4.87_{\pm 0.40}$ |
| FedProTIP (-t) | $\mathbf{28.90}_{\pm 0.39}$ | $7.46_{\pm 0.30}$ | $\mathbf{28.98}_{\pm 1.20}$ | $6.65_{\pm 0.57}$ | $\mathbf{29.32}_{\pm 0.58}$ | $5.70_{\pm 0.51}$ |
| **FedProTIP** | $\underline{28.78}_{\pm 0.08}$ | $\mathbf{1.59}_{\pm 0.20}$ | $\underline{28.06}_{\pm 0.83}$ | $\mathbf{2.10}_{\pm 0.47}$ | $\underline{25.89}_{\pm 0.77}$ | $\mathbf{1.49}_{\pm 0.32}$ |

## B  CONVERGENCE ANALYSIS

At iteration $t$, the local model of client $k$ is updated as:

$$W_k^{t+1} \leftarrow W_k^t - \eta \nabla \tilde{F}_k(W_k^t, \xi_k^t), \tag{13}$$

where $F_k$ is the local loss function computed on mini-batch $\xi_k^t$ sampled from client $k$'s local data. To mitigate interference with past tasks, the local gradients are projected onto the orthogonal complement of the subspace spanned by the core bases $\Phi$ extracted from previous tasks:

$$\nabla \tilde{F}_k(W_k^t, \xi_k^t) = \nabla F_k(W_k^t, \xi_k^t) - \Phi\Phi^\top \nabla F_k(W_k^t, \xi_k^t) = P \nabla F_k(W_k^t, \xi_k^t), \tag{14}$$

where $P = I - \Phi\Phi^\top$ is the projection matrix. Here, $P$ is an idempotent projection matrix, meaning it satisfies: $P^2 = P$. To verify this, we note that the columns of $\Phi$ are bases, hence:

$$P^2 = (I - \Phi\Phi^\top)^2 = I - 2\Phi\Phi^\top + \Phi\Phi^\top\Phi\Phi^\top = I - \Phi\Phi^\top = P. \tag{15}$$

The eigenvalues of a projection matrix are either 0 or 1, so:

$$\|P\|_2 = \max\{|\lambda_i|\} = 1. \tag{16}$$

As a result, the norm of the projected gradient is bounded as:

$$\|\nabla \tilde{F}_k(W_k^t, \xi_k^t)\|_2 = \|P \nabla F_k(W_k^t, \xi_k^t)\|_2 \le \|P\|_2 \|\nabla F_k(W_k^t, \xi_k^t)\|_2 = \|\nabla F_k(W_k^t, \xi_k^t)\|_2. \tag{17}$$

We now examine whether the theoretical assumptions used in the convergence analysis of FedAvg in non-IID scenarios (Li et al., 2019) still hold under our projected update scheme. Assumptions 1 and 2 in (Li et al., 2019), which concern the smoothness of the local loss functions and the bounded variance of stochastic gradients, are unaffected because they are properties of the objective function $F_k$ itself and not of the update mechanism. Furthermore, assumptions 3 and 4 about the bound to the gradient variance and gradient norm, respectively, remain the same as

$$\mathbb{E}[\nabla \tilde{F}_k(W_k^t, \xi_k^t)] = \mathbb{E}[(I - \Phi\Phi^\top) \nabla F_k(W_k^t, \xi_k^t)] = (I - \Phi\Phi^\top) \mathbb{E}[\nabla F_k(W_k^t, \xi_k^t)], \tag{18}$$

Table 13: Average accuracy (%) across different inference settings. FedProTIP (-t) and FedProTIP correspond to task-agnostic inference without and with task identity prediction, respectively.

| Dataset | Method | Task-Aware | Task-Agnostic |
|---|---|---|---|
| 10-split CIFAR100 | FedAvg | $38.99_{\pm 6.52}$ | $15.35_{\pm 2.82}$ |
| | GLFC | $75.26_{\pm 3.43}$ | $11.86_{\pm 2.00}$ |
| | LGA | $85.04_{\pm 3.73}$ | $14.35_{\pm 1.08}$ |
| | TARGET | $69.81_{\pm 1.39}$ | $27.55_{\pm 0.89}$ |
| | LANDER | $84.19_{\pm 1.94}$ | $37.59_{\pm 3.85}$ |
| | FOT | $82.59_{\pm 0.61}$ | $41.80_{\pm 1.12}$ |
| | **FedProTIP (-t)** | $\mathbf{86.26}_{\pm 0.27}$ | $48.41_{\pm 0.51}$ |
| | **FedProTIP** | – | $\mathbf{86.00}_{\pm 0.75}$ |
| 10-split ImageNet-R | FedAvg | $21.47_{\pm 0.11}$ | $8.15_{\pm 0.25}$ |
| | GLFC | $30.37_{\pm 3.06}$ | $3.18_{\pm 1.23}$ |
| | LGA | $41.73_{\pm 7.13}$ | $5.76_{\pm 1.70}$ |
| | TARGET | $37.64_{\pm 0.67}$ | $14.60_{\pm 0.66}$ |
| | LANDER | $52.66_{\pm 1.29}$ | $23.96_{\pm 0.67}$ |
| | FOT | $54.37_{\pm 1.35}$ | $26.31_{\pm 1.91}$ |
| | **FedProTIP (-t)** | $\mathbf{61.72}_{\pm 0.04}$ | $\mathbf{35.64}_{\pm 0.81}$ |
| | **FedProTIP** | – | $\mathbf{54.48}_{\pm 1.87}$ |
| 6-split DomainNet | FedAvg | $10.48_{\pm 0.70}$ | $10.72_{\pm 0.14}$ |
| | TARGET | $18.11_{\pm 0.30}$ | $20.62_{\pm 0.18}$ |
| | LANDER | $15.45_{\pm 0.58}$ | $21.59_{\pm 1.00}$ |
| | FOT | $26.27_{\pm 0.43}$ | $24.13_{\pm 0.25}$ |
| | **FedProTIP (-t)** | $\mathbf{28.75}_{\pm 1.45}$ | $\mathbf{28.85}_{\pm 1.46}$ |
| | **FedProTIP** | – | $25.30_{\pm 0.30}$ |

Table 14: Per-task and per-client communication cost (MB) comparison between FOT and FedProTIP in 10-split CIFAR100. FOT transmits randomized activations ('Act') at the end of each task, resulting in a constant overhead of 48 MiB per task. In contrast, FedProTIP communicates only core bases and reference vectors, whose combined size remains below 10 MB and decreases over time as fewer bases are extracted.

| Task | 1 | 2 | 3 | 4 | 5 | 6 | 7 | 8 | 9 | 10 |
|---|---|---|---|---|---|---|---|---|---|---|
| Model | 46.686 | 46.764 | 46.842 | 46.920 | 46.998 | 47.077 | 47.155 | 47.233 | 47.311 | 47.389 |
| Act | 48 | 48 | 48 | 48 | 48 | 48 | 48 | 48 | 48 | 48 |
| Bases | 9.794 | 5.646 | 2.841 | 1.990 | 1.278 | 0.693 | 0.489 | 0.434 | 0.442 | 0.264 |
| Ref.vecs | 0.000 | 0.000 | 0.000 | 0.000 | 0.000 | 0.000 | 0.000 | 0.000 | 0.000 | 0.001 |

and

$$\mathbb{E}[\nabla \tilde{F}_k(W_k^t)] = (I - \Phi\Phi^\top)\,\mathbb{E}[\nabla F_k(W_k^t)]. \tag{19}$$

In conclusion, considering the local update rule by incorporating a projection step while keeping the global aggregation identical to FedAvg, all key assumptions necessary for the convergence guarantees of FedAvg remain valid. Consequently, the convergence behavior of the algorithm is preserved.

## C EXPERIMENTAL DETAILS

### C.1 DATASETS

We evaluate our methods and baselines on 3 datasets: CIFAR100, DomainNet, and ImageNet-R. Details on number of classes and dataset division are given in Table 15.

**CIFAR100** CIFAR100 contains $32\times32$ sized images from 100 classes, with 600 images per class. In our class-incremental setting, we divide 100 classes into 10 tasks each consisting of 10 classes.

**ImageNet-R** ImageNet-R (ImageNet-Rendition) (Hendrycks et al., 2021) consists of artistic renditions of 200 object classes from ImageNet, including cartoons, graffiti, and paintings, providing a

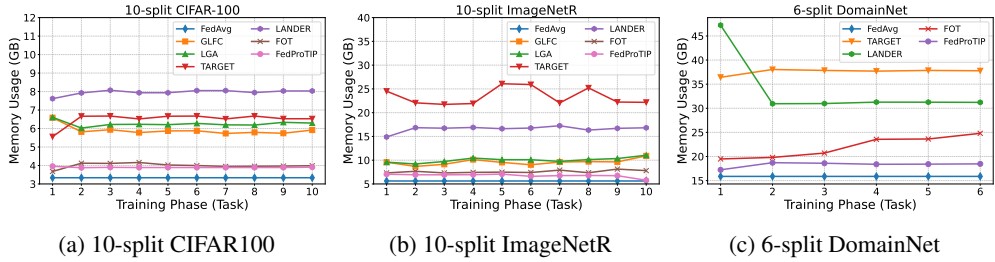

(a) 10-split CIFAR100      (b) 10-split ImageNetR      (c) 6-split DomainNet

Figure 5: GPU memory usage (GB) on a single NVIDIA H200 GPU. We report the maximum GPU memory allocated at each training phase.

Table 15: Dataset details used in experiments.

| Dataset | # Classes | # Tasks | # Train | # Test |
|---|---|---|---|---|
| CIFAR100 | 100 | 10 | 50,000 | 10,000 |
| DomainNet | 345 (per domain) | 6 | 60,000 | 20,674 |
| ImageNet-R | 200 | 5/10/20 | 67,080 | 19,464 |

benchmark for evaluating model's robustness to distribution shifts. In the class-incremental setting, we conduct experiments on ImageNet-R by dividing its 200 classes into 5, 10, and 20 tasks, with each task containing 40, 10, and 5 classes, respectively.

**DomainNet** DomainNet consists of 224×224 images spanning six visual domains: real, clipart, infograph, painting, quickdraw, and sketch, with each domain treated as a separate task. For training, we sample 10k images per domain, while evaluation uses the full test set. In the main paper, we report results using task ordering 1 (clipart → real → painting → sketch → infograph → quickdraw). For completeness, Table 11 presents results under task ordering 2 (clipart → infograph → painting → quickdraw → real → sketch).

### C.2 MODEL ARCHITECTURE

We use a ResNet-18 (He et al., 2016) pre-trained on ImageNet-1K as the backbone network for all datasets in the main paper. After learning the first task, we freeze the first two residual blocks of ResNet and only update the remaining parts of the model. At the end of each task, the parameters of the last fully connected layer are extended by adding neurons as classes are incremented. In addition, while learning new tasks we freeze the parameters of the last fully connected layer corresponding to previously learned tasks.

### C.3 TRAINING DETAILS

In all experiments, we use the SGD optimizer with a learning rate of 0.01 and a weight decay of $5 \times 10^{-4}$ for all baselines. Unless otherwise stated, the batch size is set to 64. For training, the local epoch is fixed at 5 and the number of global rounds per task is 50 for CIFAR100 and ImageNet-R, and 20 for DomainNet. To maintain consistent number of selected clients across different experiments, we apply a client fraction 1.0 at each round for 5 clients, and 0.5 and 0.25 for 10 and 20 clients, respectively. We set the threshold $\epsilon_l = 0.7$ for all datasets. Additional ablation study on the threshold value is provided in Appendix A.3. We describe training details for each baseline in the following.

**GLFC** GLFC (Dong et al., 2022) employs exemplar replay by storing a subset of raw samples for each task. For CIFAR100, following the original paper we set the memory size to 2000; to satisfy memory constraints, for DomainNet and ImageNet-R the memory size is limited to 1000. GLFC incorporates sample reconstruction optimization to select the best old model on a proxy server, where the selected model is used in the next task via distillation. For this optimization we use the L-BFGS optimizer with a learning rate of 0.5 for CIFAR100 and DomainNet, and 0.1 for ImageNet-R.

**LGA** LGA (Dong et al., 2023) extends GLFC by relying on a gradient encoding model to reconstruct perturbed images from the gradients received on a proxy server. Additionally, it introduces self-

supervised prototype augmentation to enhance selection of the best old model from the reconstructed perturbed prototype images. In our experiments, we use LeNet as the gradient encoding model for all datasets, and the SGD optimizer to generate perturbed images. We retain the same experimental settings as implemented by GLFC if the two approaches share the same configurations.

**TARGET** TARGET (Zhang et al., 2023) leverages the previously trained global model to distill knowledge from past tasks into the current model while also training a generator to produce synthetic data that captures global information from previous tasks. In our implementation, we use 8k synthetic samples with a batch size of 256 for CIFAR100, following the original paper's hyperparameters for generator training rounds, distillation schedules, and learning rates. For DomainNet, we generate 12,800 synthetic samples in batches of 64, with 200 rounds of data generation and 100 generator iterations per round. For ImageNet-R, we use 12,800 synthetic samples with a batch size of 64, and set the data generation process to 40 rounds with 40 generator iterations per round to fit GPU memory constraints.

**LANDER** LANDER (Tran et al., 2024) utilizes label text embeddings (LTE) generated by pre-trained language models as anchor points, constraining feature embeddings of the training data around the corresponding class LTEs. Additionally, these anchors guide the generator optimization, ensuring that the global model embeddings of synthetic samples remain close to LTEs, thereby generating more meaningful samples. We follow the same experimental settings and use the provided LTEs for LANDER on CIFAR100 as suggested in the original paper. For other datasets, since the official implementation does not include LTE generation, we construct the LTE pool using a pretrained CLIP model (Radford et al., 2021). We adopt the same prompt template, "A photo of a class", where `class` denotes the label of each class. For DomainNet and ImageNet-R, we match the number of synthetic samples and data generation procedure used in TARGET, while keeping all other configurations consistent with CIFAR100.

**FOT** FOT (Bakman et al., 2024) adapts GPM to the FCL setting, with key differences from Fed-ProTIP occurring at the end of each task: (i) A client transmits its input representation multiplied by a standard normal vector with a predefined sampling dimension; (ii) the randomized input representations are averaged and the core bases of the gradient subspace are extracted from these aggregated representations; and (iii) the global model parameters are updated via orthogonal projection using these bases on the server side. In our implementation, for each dataset we set the sampling dimension of the standard normal vector to five times the feature size. As for the threshold required to obtain bases from the aggregated features, we use the starting value of 0.87 with an increment of 0.01 for each new task for CIFAR100, while for DomainNet and ImageNet-R we use threshold 0.9 with an increment of 0.01.

### C.4 DATA HETEROGENEITY

To assess the impact of data heterogeneity on FCL systems, we partition dataset across clients based on the heterogeneity level controlled by the Dirichlet distribution. For an IID split, we randomly shuffle the dataset indices and divide them into equal-sized subsets, ensuring each client receives a uniform share of the dataset, independent of class labels. This ensures balanced data distribution across the clients. For a non-IID split, we control heterogeneity using the Dirichlet distribution parameterized by $\alpha$. Specifically, for each class, we sample a probability vector from $Dir(\alpha)$ to determine the proportion of data assigned to each client. We prevent empty assignments, guaranteeing that each client holds at least one sample from every class present in its assigned task. Smaller values of $\alpha$ lead to a more skewed distribution, creating more severe class imbalance across clients.

# D  DISCUSSIONS

## D.1  RELATED WORKS ON FCL

Federated continual learning (FCL) tackles the challenge of continuously learning from decentralized data while maintaining knowledge across tasks. An early approach to FCL, FedWeIT (Yoon et al., 2021), decomposes parameters into task-generic and task-specific ones, focusing on a task-incremental setting where the task ID is known during inference. More recently, TagFed (Wang et al., 2024) introduces a model extraction-based approach that mitigates forgetting by maintaining task-specific sub-networks with parameter masks, selectively updating recurring tasks while employing group-wise knowledge aggregation to cluster clients based on feature-based distillation at the server. pFedDIL (Li et al., 2025) propose a personalized federated domain-incremental learning method that estimates task correlations using an auxiliary classifier to determine whether to reuse a previous model or train a new one, with final predictions obtained through a weighted ensemble of personalized models.

In the realm of replay-based methods, CFeD (Ma et al., 2022) employs knowledge distillation enabled by a surrogate dataset made available to clients as well as the server. GLFC (Dong et al., 2022; 2023) addresses catastrophic forgetting by leveraging class-aware gradient compensation and class-semantic relation distillation, while relying on memory of old examples. The follow-up studies (Liu et al., 2023; Dai et al., 2023; Li et al., 2024c;a) reduce the size of the replay cache but remain reliant upon old samples.

To address the reliance on real data, generative model-based FCL methods have been proposed. FedCIL (Qi et al., 2023) employs a GAN with an auxiliary classifier to enable generative replay, preventing forgetting and aggregating global knowledge across clients. TARGET (Zhang et al., 2023) and MFCL (Babakniya et al., 2024) introduce data-free knowledge distillation that enables the use of synthetic examples to transfer knowledge from an old global model to client models. LANDER (Tran et al., 2024) builds on this by incorporating label text embeddings from pretrained language models as anchors, generating more meaningful samples and further improving the ability to mitigate forgetting. AF-FCL (Wuerkaixi et al., 2024) leverages feature generative replay with a normalizing flow (NF) model to estimate the probability density of generated features, enabling deliberate forgetting of biased features caused by data heterogeneity.

While many prior works have reported results in the class-incremental learning (CIL) setting, where task IDs are unknown during inference, they still exhibit substantial performance degradation compared to the relatively easier task-incremental learning (TIL) scenario. In contrast, FedProTIP enables accurate task-ID prediction for each test sample, thereby achieving near-TIL performance even under the more challenging CIL setting.

## D.2  PRIVACY CONSIDERATIONS IN FEDPROTIP

Recent work (Zhang et al., 2025) shows that sampling gradients in subspaces orthogonal to the original gradient can significantly reduce this leakage while preserving utility. Building on a similar rationale, FedProTIP avoids transmitting raw gradients by updating only in orthogonal directions to prior task subspaces and low-rank subspace bases, thereby mitigating gradient inversion risks. Moreover, our task-identity predictor relies only on reference vectors, which collapse high-dimensional projected activations into a single scalar per task-subspace norm. Compared to methods that store exemplars or train generative replay models, FedProTIP naturally reduces potential privacy leakage while remaining computationally lightweight. As a future direction, these orthogonal-projection techniques could be further combined with formal privacy frameworks, such as differential privacy or secure aggregation, to provide quantifiable guarantees beyond empirical leakage reduction.

