# OpenReview forum: "Task-Agnostic Federated Continual Learning via Replay-Free Gradient Projection"
_ICLR.cc/2026/Conference — Submitted to ICLR 2026_

### Official Review · Reviewer_mHcn · 2025-10-25

**Soundness:** 3
**Presentation:** 3
**Contribution:** 2
**Rating:** 2
**Confidence:** 2

**Summary:**

The paper provides a orthogonal gradient projection for federated continual learning to mitigates the catastrophic forgetting.

**Strengths:**

The paper conducts extensive experiments on various challenging datasets, e.g., DomainNet, ImageNet-1K.
The paper's contribution is reasonable and show some potentials. However, there are some parts in the paper are not well justified, thus, limiting the paper from showing their novelty.
I'm willing to listen to the authors' responses and give update to the reviews.

**Weaknesses:**

1. The introduction and abstract should have more significant improvement to emphasize the paper's novelty and contributions. For example, the authors claimed that they propose FedProTIP to mitigate the catastrophic forgetting. However, it is not clear what is the problem that FedProTIP can solve while others can not.
2. The authors claimed that the paper is a replay-free methods. However, it is shown in the paper that the work is replay-based. Furthermore, from my limited understandings, the paper requires a lot of memory usage to store the exemplars. The authors please refer to the questions below to discuss about the works.
3. Some statements seem to be not correct in terms of technical, which make the paper not too reliable (e.g., saying GPM is a replay-free CL, actually GPM store orthogonal gradient bases to support the training).
4. Some technical issues needed to be discussed and well-explained to make the contributions more clear.

**Questions:**

1. It is not clear why FOT incurs significant communication overhead and raises potential privacy concerns?
2. The $\Phi$ needs to be saved for all tasks. Thus, it is questioned how are the size of the $\Phi$?
3. The authors claimed that they stored the last layer activations only. However, as I checked the code, the authors used the full function of update_GPM. Furthermore, no smaller subset are used in the code as shown in L242. The authors please discuss about this?
3. The function (4) requires all of the $Phi^t$ are orthonormal for all task to work well. However, at the second task, there are first task only? How do the authors guarantee the $Phi^0$ to be orthonormal?
4. Furthermore, the $\Phi^{[0:t-1]}$ needs all of the bases to be orthonormal. It is reasonable that the entire gradients are orthonormal as used in GPM. However, by using the last activation layer, can we guarantee the orthonormal characteristics?
5. Why do we only use local core bases only? we lack the V and $\Sigma$? Can we preserve the orthonormal of the $\Phi$?
6. In L242, this is resevoir sampling. However, this one seems not proved the significant performance. Furthermore, the data transmission seems not to be justified. This one needed to be discussed carefully in the experimental evaluations.
7. How (10) can show the task alignment? This part need justification in the paper.
8. The authors claimed that FedProTIP achieves the fastest training. However, due to the questions above, it is unclear how FedProTIP can achieve the fastest computation time. The authors please discuss about this.
9. The core bases in memory usage in 5.4 should be explained why we can achieve this (refers to the previous questions).
10. The GPU usage seems not properly shown the memory efficiency of the replay memory. It is how samples are stored in the GPU for the training. The authors please consider about this carefully.

---

> ### Author Response · Authors · 2025-11-24
> **Clarifications About the Contributions and Revisions (1)**
>
> We thank the reviewer for their thoughtful feedback and efforts to improve our work. Below, we address each comment in detail:
>
> ---
> ### **1. Improvements on introduction and abstract**
> We emphasize that FedProTIP is designed to address three concrete limitations of existing approaches:
> - **Catastrophic forgetting under federated training.**
> Prior GPM-style method (e.g., FOT) first lets clients train locally without constraints and then project only the aggregated global update onto past-task subspaces at the server. This mismatch between unconstrained local updates and constrained global updates can still cause forgetting and unstable convergence. FedProTIP instead performs local projected gradient descent, where each client explicitly updates only in the orthogonal complement of the global past-task subspace. This directly enforces past-task preservation at the level of client gradients, reducing forgetting without replay, which would otherwise require computationally expensive sample storage and sampling in federated settings.
> - **Privacy-aware subspace sharing instead of activation sharing.**
> FOT requires clients to upload full intermediate activation matrices to the server, which are known to be vulnerable to feature/gradient inversion attacks. In contrast, FedProTIP never transmits raw activations. Clients (i) randomly sample and project their local activations, (ii) extract low-rank core bases via SVD, and (iii) send only these compact bases to the server. The server aggregates these bases into a global subspace, so only anonymized, low-dimensional directions are centralized, not per-sample feature vectors.
> - **Task-agnostic inference without extra models or replay.**
> Most existing FCL methods are inherently task-incremental: they assume the task ID is known at inference and route predictions through the corresponding head. FedProTIP introduces a task-identity prediction (TIP) mechanism that operates purely on the learned gradient subspaces and final-layer activations. This enables task-agnostic inference using the same model and subspaces learned during training, without training an additional router or using extra replay data.
> ---
> ### **2. Clarification on replay-free method and memory usage**
>
>
> We clarify that by *replay-free* we mean that the method does not perform repeated forward passes on stored past data or generated pseudo-samples during training, unlike replay-based approaches such as GLFC, LGA, TARGET, and LANDER. FedProTIP never reuses old examples in the optimization loop; instead, it stores compact core bases (subspaces) and constrains future gradients to be orthogonal to those bases.
> This interpretation is consistent with prior theory work. For example, Lin et al. (ICML 2023, [1]) explicitly describe orthogonal-projection–based methods as ones that “store the gradient information of old tasks and learn the new task in the orthogonal direction to old tasks without data replay,” even though these methods maintain a memory of subspace information rather than exemplars.
>
> Regarding memory, FedProTIP stores only low-rank bases $U_l^{(t)} \in \mathbb{R}^{d_l \times r_l^{(t)}}$ per layer and task, with $r_l^{(t)} \ll d_l$, leading to total overhead on the order of $\sum_l d_l r_l^{(t)}$, which is explained in Section 5.4 (L461-L475). This is substantially smaller than exemplar-based replay, which must retain full feature maps or raw images for many samples across tasks. The actual memory of core bases is presented in Table 14 in Appendix.
>
> ---
> ### **3. Communication overhead and privacy compared to FOT**
> - **Communication**: In Section 5.4 in our manuscript, we detailed communication overhead and additionally we provided the actual cost of communication overhead of FedProTIP by comparing with FOT in Table 14 of Appendix A.5.
> - **Privacy**: As reviewer hsz1 pointed out, sharing raw activations can leak privacy. FedProTIP avoids this through a two-step design. First, each client randomly samples activations $A_l^{(t)} \in \mathbb{R}^{d_l \times m^s}$ with $m^s \ll m$ from flattened feature maps (after convolution/pooling), and locally applies (randomized) SVD and subspace projection. The server receives only core bases $U_l^{(t)}$ (low-rank orthonormal directions), not per-sample feature vectors. Second, these layer-wise bases are merged across many clients into a single global subspace via further projections, which aggregates and anonymizes information. This is fundamentally different from FOT, where the server centralizes high-dimensional activations that can be directly plugged into feature/gradient inversion attacks. In our case, the server only observes a low-dimensional subspace: many different activation sets share the same span, so reconstructing individual features from $U_l^{(t)}$ alone is a highly underdetermined inverse problem.
>
> ---
> [1] Lin et al. Theory on Forgetting and Generalization of Continual Learning, ICML 2023.

---

> > ### Comment · Reviewer_mHcn · 2025-11-26
> > **Reply to authors (1)**
> >
> > I have read the response, I appreciate the authors in making the response and improving the manuscript. I will give decision after the discussion and the revised manuscript is updated. Besides, I want to have some further questions and feedback following up my comments:
> > 1. I agree that your work does not require the inference of the networks on the stored data. This will achieve better computation. However, there are two concerns coming up. First, although I agree with the author's statement, I still believe that using the phrase replay-free will make some misleading to the readers, thus, making the paper less technically sounds. Second, as the work does not use any further computation overhead via network inference, I believe that a computation complexity analysis to emphasize that the proposed algorithm is really better than that of "replay-based method".
> > 2. The Eq. (10) needed to be explained carefully to show how the task identification is conducted. Furthermore, as the new task come, we have to measure the new task via models inference. How can we guarantee that although the new task is similar to the old task, the gradients are identical. Please note that as the model progressed, the gradient trajectory become different with the previous one. The proper foundational theorem and experimental analysis are needed to prove this part in order to improve the paper's contribution and soundness.

---

> > > ### Author Response · Authors · 2025-12-03
> > > **Reply to additional comments from the reviewer (1)**
> > >
> > > ### **1. Replay-free method**
> > >
> > > We agree that the phrase replay‑free can be misunderstood if not defined precisely. In this work, we use replay in the standard continual‑learning sense of reusing stored real or synthetic examples inside the optimization loop, i.e., performing additional forward/backward passes on past data during training (as in GLFC, LGA, TARGET, LANDER, etc.). FedProTIP does not perform any such real/synthetic data replay: once a task is finished, its raw examples are never fed back into the network for gradient updates.
> > >
> > > Nonetheless, to avoid any possible confusion and improve technical soundness, we will clarify the terminology explicitly in the introduction and use wording such as “data‑replay‑free”:
> > >
> > > - We are also happy to soften the title/introduction phrasing if needed, e.g., by changing “replay‑free” to “data‑replay‑free” or “exemplar-free” gradient projection” and emphasizing that FedProTIP still stores subspace statistics rather than exemplars.
> > >
> > >
> > > We hope this makes our intended meaning clear.
> > >
> > > ---
> > > ### **2.Complexity analysis**
> > >
> > > - **Comparison to replay-based methods**:
> > > During local training at the client side, replay-based methods maintain a buffer of old examples of size $R$ and mix them with $N_t$ current-task data in every local update. Per local epoch, each client must process $(N_t + R)$ examples, so the training cost per client/device is $O((N_t+R)C)$ per epoch when $C$ is the cost of one forward-backward pass through model. In practice, keeping accuracy competitive typically requires $R$ to be on the same order as (or larger than) the current task size. FedProTIP trains only on current‑task data, so the number of forward/backward passes per epoch is the same as FedAvg. On top of this, FedProTIP adds a cheap gradient-projection step. For layer $l$ of network with gradient dimension $d_l$ and rank of stored core bases of previous tasks $R_l$, the projection costs $O(d_l R_l)$. The overall local training complexity per client is $O(N_t (C+\sum_l d_l R_l))$, which is typically dominated by standard forward-backward cost $N_tC$ as matrix multiplication in gradient-projection could be done very efficiently.
> > > At the server side, FedProTIP performs SVD on the sampled activations to extract core bases (Sec. 5.4), which is much more efficient than generative replay‑based methods, which require training and running a generator to produce pseudo‑samples for replay; empirically, this is reflected in Fig. 4, where generative replay methods are significantly slower than FedProTIP, especially on high‑resolution datasets like DomainNet and ImageNet‑R.
> > >
> > > - **Complexity during inference**:
> > > As the reviewer notes, the Task Identity Prediction (TIP) module can be implemented in a computationally efficient way. We will add the following analysis.
> > >
> > > Let $d_L$ denote the final-layer feature dimension, $T$ is the number of tasks, $K$ is the number of clients, and $r_t$ denotes the number of bases for task $t$.
> > >
> > > **(i) Precomputation of the projection matrix**: For each task $t$, we precompute the projection matrix $P_t = U_L^{(t)} U_L^{(t)\top}$ once, using the learned bases $U_L^{(t)} \in \mathbb{R}^{d_L \times r_t}$. This costs at most $O(d_L^2 r_t)$ per task and is done offline.
> > >
> > > **(ii) Reference vector computation**: For each client, each task $\tau$, and each past task $t \le \tau$, we compute $\omega^{(\tau,t)} = \lVert P_t a_L^{(\tau)}\rVert_2$. Given precomputed $P_t$, each projection costs $O(d_L^2)$, so forming all reference vectors on one client costs at most
> > > $O(T^2 d_L^2)$, and $O(K T^2 d_L^2)$ across $K$ clients. This is a one-time cost during training.
> > >
> > > **(iii) TIP at test time**: For test activation $a_L^{\text{te}}$: Computing the relevance vector $\hat{\omega} \in \mathbb{R}^T$ with entries $\hat{\omega}_t = \lVert P_t a_L^{\text{te}}\rVert_2$ requires $T$ projections, yielding the cost of $O(T d_L^2)$. Computing cosine similarities between $\hat{\omega}$ and $T$ stored reference vectors per client costs $O(K T^2)$ (dot products and norms of $T$-dimensional vectors). In summary, the total TIP overhead per test sample is $O(T d_L^2 + K T^2)$.
> > >
> > > In our experiments, the TIP step for an entire test split completes within a few seconds and adds virtually no observable overhead to inference time.

---

> > > ### Author Response · Authors · 2025-12-03
> > > **Reply to additional comments from the reviewer (2)**
> > >
> > > ### **3. Question about Eq. (10) and task identification**
> > > We clarify how Eq. (10) is used and why task identity remains recoverable even when tasks are similar and the model continues to evolve.
> > >
> > > First, the core bases $U^{(t)}$ in Eq. (10) are task‑specific incremental bases extracted during training. In Eq. (6), for each task $t$, we project its activations onto the orthogonal complement of all previous tasks’ subspaces and then perform SVD on these projected activations. As a result, $U^{(t)}$ is constructed only from the new directions that are not explained by earlier tasks $1,\dots,t-1$. Even if two tasks are semantically similar and share many features, we store only the incremental difference of task $t$ relative to previous tasks. For example, if task 0 and task 2 are very similar (but not identical), then $U^{(2)}$ contains only directions that are absent from $U^{(0)}$.
> > >
> > > Second, during training we apply gradient projection at every local update. The projected gradient is $\nabla \tilde{W}^{(t)}_k = \nabla W^{(t)}_k - \Phi^{(1:t-1)}\Phi^{(1:t-1)\top}\nabla W^{(t)}_k$, which explicitly removes components along previously learned subspaces. This means that when we train on task 2, the model preserves the gradient space of task 0 and updates the parameters only in the new directions associated with task 2, which do not overlap with task 0. In other words, even though raw gradients do drift over time, the projected gradients for each task are constrained to live in disjoint subspaces, and the bases $U^{(t)}$ are fixed once learned.
> > >
> > > TIP then does not rely on raw gradient values being identical; instead, it compares the distributional pattern of relevance across all task subspaces. Eq. (10) gives, for each task $\tau$, a reference vector $\omega^{(\tau)}$ whose entries $\omega^{(\tau,t)} = \lVert U_L^{(t)}U_L^{(t)\top}a_L^{(\tau)}\rVert_2$ measure how task $\tau$’s features align with each task’s subspace. At test time, we form a relevance vector $\hat{\omega}$ from the test activation and compare it (via cosine similarity) to the stored reference vectors. Even if two tasks are similar, their incremental subspace components differ: samples from task 2 have substantial mass in directions captured by $U_L^{(2)}$ but absent from $U_L^{(0)}$, so the resulting reference vectors $\omega^{(0)}$ and $\omega^{(2)}$ are distinguishable when the server matches $\hat{\omega}$ to the stored references.

---

> ### Author Response · Authors · 2025-11-24
> **Clarifications About the Contributions and Revisions (2)**
>
> ### **4. Size of $\Phi$**
> The server does not store separate bases per client. For each layer $l$ and task $t$, it receives client bases ${U_{l,k}^{(t)}},{k=1\cdots K}$ and aggregates them into a single global orthonormal basis $\Phi_l^{(t)}$ using Eq. (9). During this aggregation, new directions that lie in the span of the current global subspace are discarded, so the rank of the global basis grows sublinearly with the number of clients $K$. The server therefore only keeps one global basis $\Phi_l \in \mathbb{R}^{d_l \times R_l}$ per layer, not $K$ copies, and the memory cost is $O(d_l R_l)$ rather than $O\big(d_l \sum_{k,t} r_{l,k}^{(t)}\big)$. We’ll clarify this further in our revised manuscript.
>
> In Section 5.4 and Appendix A.5 (Table 14), the reported “Bases” sizes correspond to these aggregated global subspaces. For CIFAR100–ResNet18, raw activations would require 480 MB over 10 tasks, whereas the final global bases occupy only about 23.9 MB in total after aggregation over all clients.
>
> —
> ### **5. Clarification on update_GPM function**
>
> We clarify that we do not store activations at any point, neither during training nor inference. Activations are used only ephemerally: they are computed on-the-fly, fed into (randomized) SVD or reference computation, and immediately discarded.
> During training, we use activations from all layers as input to “update_GPM” function to extract per-layer core bases $U_l^{(t)}$ in the standard GPM manner. Across tasks, the only quantities that are stored are these low-rank bases $U_l^{(t)}$ (and their aggregated versions $\Phi_l$), not raw per-sample activations. By contrast, task-identity prediction (TIP) relies only on final-layer activations and the corresponding stored final-layer bases to construct reference and relevance vectors (Eq. 10–11 in Section 4.4).
> - Regarding subsampling: for all hidden layers used solely for GPM, we subsample activations with dimension $512$ before SVD. For the last layer, which directly feeds TIP, we intentionally do not subsample so that task-identity prediction is computed from the full final-layer representation without additional approximation.
>
> ---
> ### **6. Orthonormalization of core bases**
> Eq. (4) always uses an orthonormal subspace by construction. (For simplicity, we omit the layer index and denote by $\Phi^{(t)}$ the global basis after task $t$.) At task 1 and client $k$, we compute the SVD in Eq. (7), $A_{k}^{(1)} = U_{k}^{(1)} \Sigma_{k}^{(1)} V_{k}^{(1)\top}$, and the server concatenates $\{U_{k}^{(1)}\}$ over $k$ and performs a Gram–Schmidt process to obtain the global basis $\Phi^{(1)}$ with orthonormal columns.
>
> How we maintain orthonormality for later tasks. After finishing task $t$, the server updates the global subspace by merging new client bases with the existing one. Processing clients sequentially, we:
> - start from the current orthonormal global basis $\Phi^{(t)}=\Phi^{(t-1)}$
> - receive client $k$’s local core basis $U_{k}^{(t)}$ (from activations projected onto complement of previous task spaces)
> - concatenate and re-orthogonalize: $\Phi^{(t)}\leftarrow\text{orth}([\Phi^{(t)},U_{k}^{(t)}])$.
>
> After all clients for task $t$ have been processed in this way, the resulting global basis $\Phi^{(t)}$ has orthonormal columns by construction, and Eq. (4) always operates with an orthonormal projection matrix $P^{(t)}=\Phi^{(t)}\Phi^{(t)\top}$. We will clarify this orthogonalization step more explicitly in the algorithm and around Eq. (9).
>
> ---
> ### **7. Construction of core bases**
> - **We don’t use local bases for gradient projection.**
> After each task $t$, every client $k$ runs SVD $\tilde A_{k}^{(t)}=U_{k}^{(t)}\Sigma_{k}^{(t)}V_{k}^{(t)\top}$ on its sampled, projected activations and sends only the core bases (a subset of columns of $U_{k}^{(t)}$) to the server. The server aggregates these local bases across clients and tasks, orthogonalizes them, and obtains a global orthonormal basis $\Phi^{(t)}$. These global bases are then broadcast back to clients and only $\Phi^{(t)}$ is used to project local gradients during training (as in Eq. (4)).
> - **Why we only keep $U$ and not $V$.**
> For projection and TIP we only need the subspace spanned by the activations. That subspace is completely characterized by the left singular vectors: if $\tilde A^{(t)}=U^{(t)}\Sigma^{(t)}V^{(t)\top}$, then $\text{col}(\tilde A^{(t)}) = \text{col}(U^{(t)})$, and the associated projector is $U^{(t)}U^{(t)\top}$. The right singular vectors $V^{(t)}$ never appear in the projector, so we do not need to store or communicate them.
> - **How we use $\Sigma$ and preserve orthonormality.**
> We use the singular values in $\Sigma^{(t)}$ to decide how many columns of $U^{(t)}$ to keep, so we retain only the “core” directions. Once selected, these columns of $U^{(t)}$ are already orthonormal (by SVD). After aggregation at the server, we re-orthogonalize the concatenated bases to ensure that the global $\Phi^{(t)}$ remains orthonormal for all tasks.

---

> > ### Author Response · Authors · 2025-11-24
> > **Clarifications About the Contributions and Revisions (3)**
> >
> > ### **8. Performance of sampling and Data transmission**
> > - **The operation at L242** is a standard random sampling step on layer activations before applying (randomized) SVD. Instead of running SVD on the full activation matrix, we subsample a small number of columns to construct a low-dimensional sketch, which significantly reduces the decomposition cost while preserving the dominant directions of the activation space. We empirically validate that this does not harm performance: we report results for multiple sampling dimensions in Answer [Q1] to reviewer Z5rx, showing that accuracy is stable across a wide range of sampling sizes.
> > - **Data transmission from client to server.** This sampling is purely local and only used to compute the subspace; the sampled activations themselves are never transmitted. Beyond standard FL model updates, the only extra information sent from clients to the server are: (i) per-task core bases (low-rank matrices per layer) once at the end of training each task, and (ii) a single set of lightweight TIP reference vectors per client after training (dimension $T$, which is number of tasks). The total communication cost in MB, including both core bases and reference vectors, is quantified in Table 14 of Appendix A.5, where FedProTIP adds only modest overhead over FedAvg and remains significantly more communication-efficient than FOT, which transmits full activation matrices.
> > ---
> > ### **9. Task alignment**
> >
> > In Eq. (10), $\omega^{(\tau,t)} = \lVert U_L^{(t)} U_L^{(t)\top} a_L^{(\tau)}\rVert_2$ is exactly the norm of the orthogonal projection of the activation $a_L^{(\tau)}$ onto the task-$t$ subspace spanned by the orthonormal columns of $U_L^{(t)}$. In linear algebra, this projection norm is a standard measure of how much a vector “lies in” (is aligned with) a subspace: if $a_L^{(\tau)}$ is well explained by the task-$t$ features, $\omega^{(\tau,t)}$ is large; if it is nearly orthogonal, $\omega^{(\tau,t)}$ is small. We will clarify this geometric interpretation around Eq. (10) in the revised paper.
> >
> > ---
> > ### **10. Training time**
> >
> > Table in Answer 7 to reviewer Z5rx reports a per-task runtime breakdown for FedProTIP. During training, FedProTIP shares the same structure as FedAvg: it performs local projected gradient updates and standard global aggregation. The main extra cost comes from base extraction (collecting client activations, extracting local bases, and forming the global projection memory) at the end of each task. This step grows over tasks but remains on the order of at most a few hundred seconds per task, and it is executed only once at the end of each task. In practice, this is substantially faster than generative replay–based methods (e.g., TARGET, LANDER), whose reconstruction and replay pipelines dominate wall-clock time.
> >
> > ---
> > ### **11. Memory usage and GPU usage**
> > We thank the reviewer for pointing out the ambiguity between “memory usage” and “GPU memory usage” in Section 5.4, and we clarify below as well as in revised manuscript.
> > - **Core bases vs replay memory.**
> > FedProTIP stores only low-rank core bases per layer, $U_l^{(t)} \in \mathbb{R}^{d_l \times r_l^{(t)}}$ with $r_l^{(t)} \ll d_l$, so the total persistent memory footprint scales as $\mathcal{O}\big(\sum_l d_l r_l^{(t)}\big)$ per task. In contrast, replay-based methods store raw samples (or synthetic images) of shape $N_{\text{replay}} \times C \times H \times W$, which scales as $\mathcal{O}(N_{\text{replay}} C H W)$ and quickly dominates storage. Table 14 in Appendix A.5 makes this concrete: for CIFAR-100, storing raw activations or exemplars takes about $48$ MB per task, whereas the global core bases of FedProTIP remain an order of magnitude smaller (e.g., at most $9.8$ MB at early tasks and dropping below $1$ MB at later tasks), even after aggregating across clients and tasks.
> > - **GPU memory usage.**
> > The “GPU memory usage” curves in Figure 5 report peak device memory during training (model parameters, activations for a mini-batch, plus any replay batches or generative models) rather than the persistent size of replay buffers or bases on disk/host. Replay-based methods must repeatedly load replay exemplars or generator outputs onto the GPU, increasing peak memory. FedProTIP does not maintain replay batches and uses only a small set of core bases, which are applied as linear projections and can be stored or streamed efficiently. This is why FedProTIP has the lowest peak GPU memory among other FCL methods in Figure 5, even though its persistent footprint in bases is already small (Table 14). We will clarify this distinction in the revised text and explicitly separate “GPU memory during training” from “method memory”.

---

> > > ### Comment · Reviewer_mHcn · 2025-11-26
> > > **Reply to authors (3)**
> > >
> > > 1. **Sampling:** I believe that the analysis and detailed discussion still needed to be conducted. How can the sampled activations can hold the generalization of all activations?
> > > 2. **Task Alignment:** As I understand that Eq. 10 is the norm of cosine similarity between $U^{(t)}(U^{(t)})^T$ and $a^{(\tau)}$? Following up this, how can $U^{(t)}(U^{(t)})^T$ represents $a^{(t)}$? Furthermore, if we take the norm, how can we see the strong alignment between two vectors? what if the value of the multiplication are negative? I believe a good justification should be made in the revised manuscript.

---

> > > > ### Author Response · Authors · 2025-12-03
> > > > **Reply to additional comments from the reviewer (4)**
> > > >
> > > > ### **6. Sampling**
> > > > We appreciate the reviewer’s concern about whether sampled activations can faithfully represent all activations. In FedProTIP, we explicitly trade a small amount of approximation for substantial efficiency. In particular, our goal is not to preserve every individual activation, but to recover the dominant subspace of the activation matrix.
> > > >
> > > > Formally, for each layer $l$ and task $t$, we have an activation matrix $a_l^{(t)} \in \mathbb{R}^{d_l \times m}$, where columns are activations. The matrix is extremely wide and highly redundant: its rank is at most $\min{(d_l, m)}$, and the effective rank is typically much smaller. Theoretical results on randomized low‑rank approximation show that, for low‑coherence data matrices, uniform sampling in the data dimension yields a good spectral approximation of the Gram matrix [1]. In our setting, uniformly sampling activation columns and running SVD on the sampled matrix therefore provides an accurate estimate of the principal activation subspace at a fraction of the cost.
> > > >
> > > > While more sophisticated importance sampling schemes based on leverage score where each column is sampled with probability proportional to its statistical leverage can provide theoretical guarantees [1], we deliberately use uniform sampling as a simple and effective alternative, and leave leverage‑score–based sampling as future work.
> > > > We will add a short discussion in the appendix explicitly stating that (i) we use SVD on sampled activations to balance accuracy and efficiency, and (ii) due to the high redundancy and low effective rank of the activation matrices, a relatively small random subset of activations is sufficient to approximate the dominant subspaces that FedProTIP needs.
> > > >
> > > > [1] Cohen et al. "Uniform sampling for matrix approximation." Proceedings of the 2015 conference on innovations in theoretical computer science. 2015.
> > > >
> > > > ---
> > > > ### **7. Task alignment**
> > > > Eq. (10) is not a cosine similarity. It is defined as $\omega^{(\tau,t)} = \big \lVert U_L^{(t)} U_L^{(t)\top} a_L^{(\tau)} \big \rVert_2$, where $U_L^{(t)} \in \mathbb{R}^{d_L \times r_t}$ has orthonormal columns that span the feature subspace of task $t$. When $U$ has orthonormal columns, the matrix $P = U U^\top$ is the orthogonal projection matrix onto the column space of $U$ (please see [2]). Thus, Eq. (10) measures the $\ell_2$ norm of the activation projected onto the task-$t$ subspace via the projector $U_L^{(t)} U_L^{(t)\top}$, i.e., how much of the test activation lies in that subspace.
> > > >
> > > > Cosine similarity appears later, only at the server side, when we compare:
> > > > the relevance vector $\hat{\omega}$ computed from a test activation, and the stored reference vectors $\omega_k^{(t)}$ computed from training activations on each client.
> > > > Specifically, we use cosine similarity between these vectors of projection norms to infer the most likely task identity. We will revise the text around Eq. (10) to make this distinction explicit: Eq. (10) defines a projection-based relevance score, while cosine similarity is used afterwards to compare relevance patterns between training and test samples.
> > > >
> > > > [2] Saha et al., “Gradient Projection Memory for Continual Learning,” ICLR 2021

---

> > ### Comment · Reviewer_mHcn · 2025-11-26
> > **Response to authors (2)**
> >
> > 1. **Size of Phi:** I agree that the method (a) only requires T core bases on the memory of every clients. Although there are concerns about the task gradient similarities (in which I have asked above), here I only want to discuss about the memory efficiency. I agree that the ResNet-18 over 10 tasks on FOT requires 480 MB is due to 10 (tasks) x 48 MB (size of ResNet-18). How can we achieve 20 times better in terms of compression rate. I believe that if you can make a raw estimation at least in the appendix, it would much improve the contribution's novelty.
> > 2. **Last layer activation**: I'm sorry that my concern still remained. From my limited understanding, I will make a simple math formula to clarify my question. For example, there are two vectors $A = [a_{1}, a_{2}, ... , a_{d-k}, ..., a_{d}]$ and $B = [b_{1}, b_{2}, ... , b_{d-k}, ..., b_{d}]$. If two vectors are orthogonal, we should have $A \cdot B = 0$, which means $\sum^{d}\_{i=1} a_i b_i = 0$. However, there is not guarantee that $\sum^{d}\_{i=d-k} a_i b_i = 0$. Can you please explain this for me?

---

> > > ### Author Response · Authors · 2025-12-03
> > > **Reply to additional comments from the reviewer (3)**
> > >
> > > ### **4. Memory efficiency regarding core bases**
> > > For each layer $l$, let $d_l$ be the (flattened) feature dimension.
> > > - **FOT**: For each task $t$ and layer $l$, FOT stores a sampled activation matrix $A_l^{(t)} \in \mathbb{R}^{d_l \times m_l}$, where in our setup $m_l = 5 d_l$. Thus, the per‑task server memory is $\sum_l d_l m_l = \sum_l 5 d_l^2$, and over $T$ tasks the total server memory scales as $\text{Mem}_\text{FOT} = O(T \sum_l 5 d_l^2)$.
> > >
> > >
> > > - **FedProTIP**: In contrast, FedProTIP does not store all sampled activations. After $T$ tasks, each layer $l$ has a global core basis $\Phi_l \in \mathbb{R}^{d_l \times R_l}$, where $R_l=\sum_{t=1}^T r_l^{(t)}$ is the final rank accumulated across tasks. The server stores only these bases, with total memory $\text{Mem}_\text{FedProTIP} = O(\sum_l d_l R_l)$.
> > >
> > >
> > > To make this concrete, for CIFAR‑100 with ResNet‑18 we report the final ranks $R_l$ and layer dimensions $d_l$ of the deeper layers:
> > > |  |$R_l$|$d_l$|
> > > |-|-|-|
> > > | layer3.0.conv1 |162|1152|
> > > | layer3.0.conv2 |586|2304|
> > > | layer3.0.downsample.0 |34|128|
> > > | layer3.1.conv1 |334|2304|
> > > | layer3.1.conv2 |771|2304|
> > > | layer4.0.conv1 |429|2304|
> > > | layer4.0.conv2 |1907|4608|
> > > | layer4.0.downsample.0 |70|256|
> > > | layer4.1.conv1 |1899|4608|
> > > | layer4.1.conv2 |1719|4608|
> > > | fc  |230|2048|
> > >
> > > Across these layers, the average fraction of dimensions kept in the basis is about $R_l / d_l \approx 0.26$, i.e., in the final model we retain roughly 26% of the full layer dimension. Using the formulas above, the per‑layer memory ratio (FedProTIP/FOT) is $\approx (d_l R_l)/(5 d_l^2) =R_l/(5 d_l) $.
> > > Plugging in the empirical average $R_l / d_l \approx 0.26$ gives the ratio $0.26 / 5 \approx 0.05$, meaning FedProTIP uses only about $5$% of the server memory that FOT requires, i.e., roughly a $20\times$ reduction.
> > >
> > > We will add this with Table 14 to make the memory advantage of $\Phi$ over FOT’s stored activations more transparent.
> > >
> > > ---
> > > ### **5. Clarification on orthogonality**
> > > Starting from the first task, we keep only the top–$r$ bases from $U^{(1)}$, which span a subspace of the full gradient/feature space, not the entire space. After learning the second task, we add new bases $U^{(2)}$ that are constructed in the orthogonal complement of the subspace spanned by $U^{(1)}$. This means we guarantee orthogonality between the stored subspaces: the columns of $U^{(1)}$ and $U^{(2)}$ are orthogonal. We do not claim that the full gradient spaces of task 1 and task 2 are orthogonal; only their retained low‑rank subspaces are.
> > >
> > > This “incomplete” orthogonality is actually intentional and beneficial for stability–plasticity, as in GPM. By keeping only a small rank per task, we preserve the most important directions for old tasks (stability), while leaving the remaining directions free so new tasks can still adapt the model (plasticity). In other words, we enforce orthogonality where it matters most (along the stored bases $U^{(t)}$), and allow overlap in the less informative directions to avoid over‑constraining future tasks.

---

### Official Review · Reviewer_hsz1 · 2025-10-28

**Soundness:** 2
**Presentation:** 3
**Contribution:** 2
**Rating:** 4
**Confidence:** 4

**Summary:**

The paper proposes an approach for FCL to mitigate the catastrophic forgetting problem. The paper extends the ideas introduced in GPM to federated setting where gradients are projected onto the orthogonal complement of the subspace spanned by prior task activations.

    1. Replay based:  Store old examples, risking privacy or exceeding storage limits
    2. Generation based: require server-side generative models, slowing aggregation
    3. Regularization based: more computational overhead

**Strengths:**

The paper is a straightforward extension of the gradient projection memory technique that was proposed for centralized continual learning.

The paper is clearly written. The gaps in existing FCL approaches were clearly identified, motivation is justified as to why a federated way to do GPM makes sense. Solution developed is explained well, claims not overstated. Experimental results are comprehensive and covers the usual FCL setting very well.

**Weaknesses:**

- The calculation of the global core base is done sequentially. Since projection is not commutative, this can favor certain clients over the others. Especially, since client 1 is always chosen first, it can be seriously advantaged over other clients.
        - My second concern is that suppressing gradient updates along core bases would likely slow down convergence when data heterogeneity is low. Specifically, new updates may not contribute much if their data distributions are similar. Therefore, will the solution be applied to only when the data distribution is expected to change across clients?
        - The paper argues that sharing the raw activations leaks privacy (absolutely true). However, there is non negligible privacy leakage from sharing the orthonormal bases from each client. In the least, it helps in identifying which client has shared and not shared bases and may even help in inferring data statistics.  Refer to -> “Subspace Leakage in Federated Representation Learning” (ICML 2022 Workshop)”, “Federated PCA with Privacy” (NeurIPS 2020).
        - Some aspects such as each client experiencing same sequence of tasks, while used in other prior art, is somewhat unrealistic
        - The idea in the paper is sound when compared to other federated continual learning, especially in not causing a significant additional overhead that exist in other baselines. However, the novelty seems to be incremental when compared to the GPM technique. The key contribution seems to be federating the computation of the core bases and merging them. Merging sequentially has the above 2 challenges, which are not addressed in the convergence analysis. Privacy issues might still remain (albeit reduced significantly).

**Questions:**

Can you clarify how the sequence of calculating the global core bases affect fairness (favoring some tasks over others purely based on the update sequence)?

Clarify whether the gradient direction suppression only happens when there is task heterogeneity.

Privacy impact (see comments above). The paper argues that existing work leaks privacy (e.g., sharing raw activations), but ends up sharing the orthonormal bases from each client. So please comment on the non-negligible privacy leakage for the proposed approach.

---

> ### Author Response · Authors · 2025-11-21
> **Clarifications About the Contributions and Revisions (1)**
>
> We thank the reviewer for their thoughtful feedback to improve our work. Below, we address each comment in detail:
>
> ---
> ### **1. Aggregation of core bases**
>
> - **What we are actually constructing.**
> Conceptually, the global core subspace at a given task is defined as the span of the local subspaces from all participating clients in that round. This is the “sum of subspaces” $S_\text{global} = span(S₁ \cup \cdots \cup S_K)$, which is commutative and symmetric in the clients: adding client 1 then client 2 or client 2 then client 1 yields the same subspace $S_\text{global}$. The sequential procedure in the paper (starting from one client’s bases and then adding orthogonalized components from the others) is just a Gram–Schmidt style implementation of an orthonormal basis of this union. While the specific basis vectors produced by Gram–Schmidt depend on the order, the subspace they span does not, as long as all clients’ bases are processed.
> - **Why client 1 is not “privileged” in practice.**
> We appreciate the reviewer’s concern regarding the ordering of clients during the construction of the global core bases. However, while the aggregation is sequential, the procedure is not inherently biased toward earlier clients. Specifically, the first client provides initial bases $S_1$, and subsequent clients contribute directions that are linearly independent of what has already been captured. This does not create favoritism but rather prevents redundancy; if two clients learn similar representations, only one representative direction is needed. Moreover, if any “later” client has a unique feature direction that is not captured by $S_1$, that direction is added regardless how “late” is that client processed.
>
> - **Empirical fairness across clients.** We evaluated Jain’s fairness index (JFI; higher is more fair, with 1 corresponding to perfect fairness) [1, 2] on per-client performance in a 5-client, 10-task CIFAR100 (iid) setting. At each task, we compute JFI over the vector of per-client accuracies and the vector of per-client forgetting (FT).
>
> | CIFAR100        | T2     | T3     | T4     | T5     | T6     | T7     | T8     | T9     | T10    |
> |-----------|--------|--------|--------|--------|--------|--------|--------|--------|--------|
> | JFI (Acc) | 0.9999 | 0.9998 | 0.9997 | 0.9997 | 0.9993 | 0.9997 | 0.9999 | 0.9999 | 0.9998 |
> | JFI (FT)  | 0.9935 | 0.9952 | 0.9953 | 0.9971 | 0.9940 | 0.9976 | 0.9994 | 0.9992 | 0.9992 |
>
> We will clarify this in the revised manuscript and are happy to include ablation evidence if desired.
>
> ---
> ### **2. Convergence under local gradient projection**
>
> Our method applies a projected local gradient that modifies only the direction (not the magnitude) of the local gradient, using a projector derived from a global subspace formed by past tasks. As shown in Appendix B, the convergence guarantees for FedAvg extend to our update rule and so the convergence behavior with respect to the current-task objective is preserved. Importantly, when client data distributions are similar (e.g., near-IID), the extracted subspaces across the clients tend to be highly overlapping. As a result, the global projector spans a low-dimensional subspace while its orthogonal complement (i.e., the subspace in which the updates happen) remains large. In such cases, the projection operator tends to remove very little of the useful signal, which implies that the convergence is not meaningfully slowed down.
>
> Please note that our projection step does not increase client drift. We'll add Lemma 1 in our revised manuscript (Appendix B) that formally shows that the variance of the projected gradients across clients is never higher than the variance before projection. This means that (even under heterogeneous settings) our update rule is stable.
>
> Finally, our experiments across a range of Dirichlet $\alpha$ values further confirm that FedProTIP provides robust performance under both low and high heterogeneity (see Table 1).
>
> ---
> [1] Li et al, Fair Resource Allocation in Federated Learning, ICLR 2020.
>
> [2] Jain et al, A quantitative measure of fairness and discrimination, Eastern Research Laboratory, Digital Equipment Corporation, 1984.

---

> > ### Author Response · Authors · 2025-11-21
> > **Clarifications About the Contributions and Revisions (2)**
> >
> > ### **3. Privacy leakage from subspace**
> >
> > We appreciate the reviewer’s point that sharing orthonormal bases can, in principle, leak information about client data. In FedProTIP, we aim to reduce (rather than eliminate) this risk compared to activation or raw data sharing. Specifically, each client first samples a small subset of activations $A_l^{(t)} \in \mathbb{R}^{d_l \times m^s}$ with $m^s \ll m$ from flattened feature maps, and then applies local randomized SVD and subspace projection. The server receives only the low-rank orthonormal directions $U_l^{(t)}$, not per-sample features or gradients.
> >
> > Since many distinct activation sets yield the same subspace, reconstructing individual examples from $U_l^{(t)}$ alone is a highly underdetermined inverse problem. Furthermore, the subsequent aggregation/orthogonalization steps further entangle client contributions, rendering the recovery even more challenging. This is in contrast with FOT-style activation sharing, where high-dimensional client features are centralized and therefore vulnerable to known feature/gradient inversion attacks.
> > That said, we agree that subspaces themselves may still encode coarse data statistics and do not claim formal privacy guarantees. Our design is best viewed as complementary to formal techniques such as differential privacy or secure aggregation, which we clarify in the revised manuscript.
> >
> > Finally, we were unable to locate the exact reference titled “Subspace Leakage in Federated Representation Learning (ICML 2022 Workshop)” and would appreciate it if the reviewer could provide the precise citation so that we can appropriately refer to it.
> >
> > ---
> > ### **4. Overlapping tasks**
> > Please refer to our answers [Q8] to Reviewer Z5rx

---

### Official Review · Reviewer_Z5rx · 2025-10-31

**Soundness:** 3
**Presentation:** 3
**Contribution:** 1
**Rating:** 2
**Confidence:** 4

**Summary:**

FedProTIP is a federated continual learning framework that mitigates catastrophic forgetting through subspace-based gradient projection and task identity prediction. It extends gradient projection methods like GPM to the federated setting without requiring task IDs, replay buffers, or generative models. The approach is validated on CIFAR100, ImageNet-R, and DomainNet, showing gains over existing FCL baselines.

**Strengths:**

* Introduces a subspace projection mechanism that operates locally on clients, avoiding privacy leaks and communication overheads associated with transmitting activations (like in FOT)
* Task-agnostic inference: a task identity prediction component that uses subspace relevance vectors.
* Avoids replay, generative modeling, and task labels, which are common barriers in existing CL techniques.

**Weaknesses:**

The comments are provided in the decreasing order of importance:
1. Novelty: The novelty in this work is very low. Almost the entire technique resembles FOT, except for two steps towards the end. FOT transmits activations to the server, while here, they only transmit the subspaces. Thus, theres a fundamental tradeoff with complexity. There is also task-agnosity during the inference time, which is novel compared to FOT.
2. The paper claims task-agnostic federated continual learning, but this claim holds only at the inference stage, not during training. During training, task boundaries and task identifiers are known. Each new task is introduced sequentially, and the model (both FedProTIP and baselines) is informed that it is now training on a particular task ID. This means the system is task-aware during optimization, subspace extraction, and gradient projection. True task-agnostic continual learning would require the model to detect task boundaries automatically or operate under mixed unlabeled task streams, which is not the case here (e.g. BGD is a task agnostic setting). Most of the baselines (GPM, FOT, LANDER) were originally designed for task-incremental settings, where both training and inference use task IDs to select the appropriate classifier head. In this paper, those methods appear to have been trained (required more clarity regarding this) in their standard task-incremental mode (each task having its own head, with task ID known during training), but then evaluated in a task-agnostic manner at test time, i.e., without providing the task ID and by concatenating all task heads into a single unified classifier. This introduces a training-testing mismatch for the baselines: During training, they optimized separate heads per task, learning disjoint class boundaries. During testing, those separate heads are simply merged, without retraining tasks. It forces task-incremental models, which were never trained for shared label spaces, to operate in a task-agnostic evaluation regime for which they were not designed. In short, I think the evaluation setup disadvantages baselines, as they are tested outside their intended framework. I suggest the baselines be trained over a more consistent setup, or the proposed method be compared against truly task-agnostic algorithms. Some recent baselines with theoretical guarantees (CFLAG, AISTATS 2025) is not cited.
3. Another drawback is that there is no rigorous convergence theorem under FL conditions; no explicit assumptions are stated about client sampling, learning rates, or optimizer. Projection changes the geometry of local updates; it may reduce alignment across clients and remove useful gradient components, both affect global convergence. The method aims to bound forgetting, but there’s no formal theoretical bound on final loss or forgetting. Without formal bounds or at least derivations, it’s unclear when/why local projected updates converge.
4. Task identity inference vulnerable to client heterogeneity and small-task regimes: TIP uses client-stored reference vectors and majority votes at the server. But (a) under high non-IID (very different client data) or when tasks are highly similar (e.g., many small classes per task), reference vectors may not generalize; (b) majority voting can be biased if only a subset of clients are representative or if malicious clients exist. The experiments hint at weaker TIP performance in 20-split ImageNet-R (they report TIP less effective there); this shows TIP may break when tasks are small or under severe heterogeneity. The paper does not analyze failure modes or robustness of TIP to skewed client distributions or adversarial/malicious voters.
5. The task-identity inference mechanism assumes that reference vectors for different tasks are distinguishable. If two tasks yield highly similar subspace relevance patterns, clients may produce nearly identical reference vectors, leading to incorrect task routing. No mechanism or experiment tests this case.
6.  Repeatedly appending orthogonal bases across many tasks can exhaust the representational capacity as the orthogonal complement shrinks. The paper lacks experiments or mechanisms showing how capacity is preserved, and experiments on sensitivity to small local datasets, which can lead to bad SVD estimates.
7. Comments on experimental section:Partial participation experiments are missing. Uses pretrained ResNeT-18 backbones and 32x32 inputs for ImageNet-R which simplifies the problem - do the baselines also use pretrained networks?  For large models (e.g., ViTs), subspace dimensionality may explode; this scalability aspect is not explored. Ablation on task identity prediction accuracy will be useful to gain more insights into the strategy. (eg. confusion matrix or precision of the predicted task IDs.)
8. Communication & Computational Complexity Not Properly Quantified - The claim of “compact core bases” lacks concrete quantification. There is no detailed accounting of how communication cost scales with model depth L, task count T, or clients K. Each client transmits (a) model updates, (b) layer-wise subspace bases, and (c) reference vectors, which increases uplink rounds beyond standard FL. Additionally, randomized SVD per layer is computationally expensive on edge devices, yet no runtime or energy analysis is provided.
9. Extreme Client Heterogeneity Not Explored: Experiments include only moderate non-IID settings. The paper does not test FedProTIP under severe heterogeneity (e.g., highly skewed label or feature distributions) or when only a subset of clients participates per task. Since subspace alignment and voting rely on similarity across clients, extreme heterogeneity could significantly degrade performance.

**Questions:**

The authors must clarify the points provided in weaknesses. In addition, these are the questions:

-How sensitive is FedProTIP’s performance to the quality and dimensionality of the extracted subspaces (from randomized SVD)? If subspace estimation is inaccurate (e.g., due to small local datasets or noisy activations), could projections remove useful gradient components?
-As the number of tasks grows, the orthogonal complement shrinks, reducing the feasible gradient space. How does FedProTIP prevent capacity exhaustion or representation stagnation over many tasks? Does the model ever discard or merge old subspaces to recover capacity?
-Projecting gradients onto orthogonal subspaces can introduce optimization instability if the projection matrix is ill-conditioned or changes abruptly between rounds. Is there any smoothing or regularization applied to mitigate oscillations between projected and unprojected directions? Does the projection step conflict with local optimizer dynamics (e.g., Adam momentum or adaptive learning rates)?
-The method mentions layer-wise subspace projection. Does this apply equally to all layers? Lower layers might benefit from more stability, while upper layers might require more flexibility. Is there an adaptive layer weighting strategy?
-While FedProTIP reduces communication by sending compact subspace bases, how much compression is achieved in practice compared to FOT’s embedding uploads? Are there quantitative results showing communication savings?
-How does FedProTIP ensure cross-client alignment of subspaces learned from heterogeneous data distributions?
-Since the server constructs projection matrices, does this not reintroduce partial centralization of feature statistics, similar in spirit to FOT, though lower-dimensional? What happens if subspace information itself leaks sensitive feature structure?
-As more subspaces are accumulated, task identity prediction could become computationally expensive (multiple alignments per test sample). What is the complexity per inference? Is it feasible for edge devices?
-The paper claims strong task-agnostic performance,  but do the benchmarks include truly overlapping or ambiguous task boundaries (e.g., same-class drift or domain shift)? If not, the effectiveness in real-world task-agnostic FL might be overstated.

---

> ### Author Response · Authors · 2025-11-22
> **Clarifications About the Contributions and Revisions (1)**
>
> We thank the reviewer for their thoughtful feedback and efforts to help us improve our work. Below, we address each comment in detail:
>
> ---
> ### **1. Novelty compared to FOT**
> We appreciate the reviewer’s concern but respectfully disagree with the assessment that the innovative aspects of FedProTIP are merely incremental. As our empirical results demonstrate, the combination of proposed innovations in gradient projections and task inference enable major advancements in performance, scalability, and privacy protection of FCL systems. In particular:
> - **Local vs. Global projection**: FOT applies gradient projections globally, which is costly from the communications perspective and raises privacy concerns. FedProTIP, on the other hand, performs projections locally by having each client extract core bases from their own activation subspaces via randomized SVD, and sends the resulting low-rank bases to the server. This provides significant performance benefits even without the task identity prediction (TIP).
> - **Task Identity Prediction**: A major limitation of FOT is that it requires task IDs at inference time. TIP in FedProTIP eliminates the need for such side information, enabling precise routing of inference problem instances without explicit labels. To our knowledge, this is the first such mechanism in FCL.
> - **Communication and Computational cost**: As a result of communicating compact bases, FedProTIP significantly reduces the communications relative to FOT (please see Table 14 in the appendix). Moreover, the use of randomized SVD significantly reduces computations, allowing faster training and improving scalability.
> - **Empirical results show performance gap**: FedProTIP significantly outperforms FOT, both in task-aware/agnostic inference scenarios. As an illustration, consider the results on CIFAR100 with 20 clients shown in Table 3; FedProTIP improves task-agnostic accuracy from 37.35% (by FOT) to 81.10%, while also reducing forgetting.
>
> ---
> ### **2. Task-agnostic CL setup**
>
> Our use of “task-agnostic” term is indeed specific to the inference regime: at test time, task ID is not provided and the model must operate over a shared label space, which is precisely the setting considered in many class-incremental learning (CIL) works. As formalized in [1], CIL naturally decomposes into: (i) within-task prediction (task-aware training) and (ii) TIP (task-agnostic inference). In that sense, our training protocol is fully consistent with CIL literature: the model is task-aware during training, but task-agnostic during evaluation. Our main contribution is precisely in addressing this mismatch between task-aware training and task-agnostic testing, and the ability to operate without side information about task IDs at inference differentiates our work from the existing methods.
>
> To further address this concern, we also trained certain baselines using unified heads, i.e., the last FC layer updated jointly across tasks. We note that CIL works (GLFC, LGA) are already trained this way, following their implementation. In the table below, FedProTIP still demonstrates the effectiveness.
>
> | ACC (%)       | Task-Aware | Task-Agnostic | w/ TIP    |
> |---------------|------------|---------------|-----------|
> | LANDER        | 82.30      | 35.83         | −          |
> | FOT           | 83.45      | 39.83         | −          |
> | **FedProTIP** | **86.72**  | **45.11**     | **86.72** |
>
> | FT (%)        | Task-Aware | Task-Agnostic | w/ TIP   |
> |---------------|------------|---------------|----------|
> | LANDER        | 3.34       | 19.24         | −         |
> | FOT           | 3.50       | 27.74         | −         |
> | **FedProTIP** | **1.85**   | **18.00**     | **1.50** |
>
> Finally, we appreciate the reviewer bringing the AISTATS reference to our attention, we are adding it to the manuscript (also see our points below).
>
> ---
> ### **3. Convergence analysis**
> The two key assumptions for convergence analysis in C-FLAG [2] are:
> - a bounded bias between true past gradient and memory gradient
> - a bound on memory gradient in terms of current-task gradient.
>
> These conditions can be quite strong in FL, where memory gradients must approximate full past-task gradients using a buffer. In ours, the analogous assumptions can be phrased in terms of projected gradients used by FedProTIP. We may assume a bounded bias between true past gradient and projected current-task gradient onto the past subspace, i.e., between $\nabla f(x_t)$ and $\Phi^{(1:t-1)} (\Phi^{(1:t-1)})^{\top} \nabla g(x_t)$. Likewise, we assume a bounded relationship between past gradient and projected current-task gradient. Under these, it appears that the main steps of C-FLAG can readily be adapted to FedProTIP to obtain analogous nonconvex convergence guarantees.
>
> ---
> [1] Kim et al, A Theoretical Study on Solving Continual Learning, NeurIPS 2022.
>
> [2] Keshri et al, On the Convergence of Continual Federated Learning Using Incrementally Aggregated Gradients, AISTATS 2025.

---

> > ### Author Response · Authors · 2025-11-22
> > **Clarifications About the Contributions and Revisions (2)**
> >
> > ### **4.TIP vulnerability**
> >
> > - **Severe heterogeneity**: TIP operates in the global feature space defined by the shared model, not in client-specific spaces. Each client generates reference vectors using activations of the same global model, so heterogeneity only affects which samples are observed but not the representation space itself. This helps reduce misalignment and promotes consistency across clients.
> > - **Similar tasks**: TIP does not rely on raw activations but on their projections onto orthogonal per-task subspaces, i.e., $\lVert U^t (U^t)\top a^\tau \rVert_2$. Each $U^t$ is constructed after removing components aligned with earlier tasks’ bases. This orthogonalization encourages reference vectors to emphasize task-specific directions even when tasks are related.
> > - **Regarding the number of tasks**: While we appreciate the reviewer’s concerns, Table 7 clearly shows that TIP maintains high task-ID accuracy for a small/moderate number of tasks. We agree that performance degrades in the 20-task ImageNet-R setting, which is expected due to limited data per task and more frequent overlaps. We will clarify this limitation and add further analysis on TIP accuracy and subspace similarity as a function of the number of tasks.
> > - **Biased majority voting**: TIP uses majority voting to aggregate task predictions that are independently computed by each client in the shared global feature space. Under the standard assumption of honest but heterogeneous clients, each vote is an unbiased signal and the result of voting provides global consensus rather than introducing any new bias.
> > - **The adversarial / Byzantine case** brought up by the reviewer (malicious clients) is not modeled in our experiments. However, the TIP mechanism could be readily combined with robust FL aggregation schemes [3,4] to handle such threats. We will clarify this assumption and discuss these robust variants in the revised manuscript.
> >
> > ---
> > ### **5. Capacity of core bases and Sensitivity to small local datasets**
> > The “core” bases extracted from early tasks already capture features that can be reused by later tasks, so the residual subspaces for new tasks are low-rank and often lie largely in the span of the existing global subspace. We then apply an SVD-based truncation with a fixed threshold criterion, so only the directions whose singular values exceed this threshold are added while the weak or noisy directions are discarded. As a result, the effective subspace dimension quickly saturates and the memory footprint remains bounded rather than growing linearly with the number of tasks. While no projection-based method can support an unlimited number of linearly independent tasks, FedProTIP delivers stable representational capacity in realistic settings.
> >
> > Empirically (please see Table 14 in Appendix), we observe that bases extracted from the first few tasks already capture most of the high-energy directions. Later tasks contribute only low-rank residuals that lie largely in the span of the current global subspace and thus contribute only marginally. To directly address the concerns about unstable subspace estimates under limited local data, we additionally ran a sensitivity analysis where we reduce the size of local datasets on Task 2 to a fraction $p \in \{1.0, 0.5, 0.25\}$, keeping all other settings fixed. Across these scenarios, the accuracy remains stable, indicating robustness of the subspace extraction process even under data constraints.
> >
> > | Task 2 data portion | $\text{acc}_{t=2}^{T=10}$ (%)  | | ACC (%) | | $\text{TIP acc}_{t=2}^{T=10}$ (%) | TIP ACC (%) |
> > |---------------------|----------------|----------------|--------|---------|-----------------------------------|-------------|
> > | | Task-Aware | Task-Agnostic | Task-Aware | Task-Agnostic | | |
> > | p = 1.0             | 79.7 |  79.7 | 82.98 | 82.63 | 100 | 99.38 |
> > | p = 0.5             | 78.8 | 78.8 | 82.54 |  82.54 | 100 | 100 |
> > | p = 0.25           | 76.8  | 72.0                   | 84.92 | 82.99   | 93.75                               | 97.50       |
> >
> > We will also include in the revised manuscript the **singular value spectrum** for different local data fractions $p$, as well as the table showing the deviation metric $D^{(p)}=\lVert diag(\Sigma^{(p)}) - diag(\Sigma^{(p=1)})\rVert_2 / \lVert diag(\Sigma^{(p=1)}) \rVert_2$ that quantifies the stability of subspace estimation.
> >
> > | $D^{(p)}$ (%) | 0.25 | 0.5 | 0.75 |
> > | --- | --- | --- | --- |
> > | layer3.0.conv2 | 1.35 | 4.50 | 0.96 |
> > | layer3.0.downsample.0 | 9.57 | 2.98 | 9.15 |
> > | layer3.1.conv1 | 1.19 | 0.58 | 1.35 |
> > | layer3.1.conv2 | 1.42 | 1.23 | 0.97 |
> > | layer4.0.conv1 | 2.76 | 1.21 | 1.61 |
> > | layer4.0.conv2 | 2.12 | 0.96 | 1.57 |
> > | layer4.0.downsample.0 | 12.10 | 6.55 | 12.72 |
> > | layer4.1.conv1 | 3.83 | 2.82 | 4.41 |
> > | layer4.1.conv2 | 1.45 | 5.13 | 1.20 |
> > | **Average** | 3.98 | 2.89 | 3.77 |

---

> > > ### Author Response · Authors · 2025-11-22
> > > **Clarifications About the Contributions and Revisions (3)**
> > >
> > > ### **6. Comments on experimental section**
> > >
> > > - **Partial participation**: We respectfully note that our experiments already include results under partial client participation. Specifically, Table 3 presents accuracy and forgetting for different numbers of clients. There, we sample only 5 clients at each global round regardless of the total number of clients. FedProTIP maintains strong performance and outperforms baselines, showing robustness to client participation rates.
> > > - **Pre-trained model**: All models (ours as well as the baselines) use the same pretrained backbone. We report results with ViT-B/16 pretrained on ImageNet as well as ResNet18 trained from scratch in the Appendix (Table 5, 6). Also, we clarify that we used 224x224 inputs for ImageNet-R.
> > >
> > > - **Confusion matrix of TIP**: We thank the reviewer for the suggestion and will include confusion matrices for TIP in the revised manuscript.
> > >
> > > ---
> > > ### **7. Communication & computational complexity**
> > >
> > > Our revised manuscript presents detailed complexity analysis of the communication and computational costs, as well as the runtime analysis. For the latter, we report the average runtime per task on 10-split CIFAR100 as shown below, where it can be seen that the “Base Extract” stage (including SVD) takes at most a few hundred seconds on a single GPU per task.
> > >
> > > | Avg. Time (sec) | Training (s) | Base Extract (s) | Extract Ref. (s) | TIP (s) | Total (s) |
> > > |------|--------------|----------------|------------------|----------------|-----------|
> > > |   | 354.86       | 449.79          | 31.17           | 6.57           | 842.38    |
> > >
> > > ---
> > > ### **8.Extreme client heterogeneity**
> > > To address this concern, we conducted experiments under the Dirichlet partition with $\alpha = 0.05$ (10-split CIFAR100). The results are shown below (also included in the revised manuscript).
> > > | ACC (%)       | Task-Aware | Task-Agnostic | w/ TIP    |
> > > |---------------|------------|---------------|-----------|
> > > | LANDER        | 57.37     | 16.22              | –          |
> > > | FOT           | 67.68           | 26.87              | –          |
> > > | **FedProTIP** | **71.20**  | **30.17**     | **70.70** |
> > >
> > > | FT (%)        | Task-Aware | Task-Agnostic | w/ TIP   |
> > > |---------------|------------|---------------|----------|
> > > | LANDER        | 4.72           |  14.85             | –         |
> > > | FOT           |  4.21          | 19.06              | –         |
> > > | **FedProTIP** | **2.04**   | **14.35**     | **0.53** |
> > >
> > > —
> > >
> > > [3] Blanchard et al. Machine learning with adversaries: Byzantine tolerant gradient descent. NeurIPS 2017.
> > >
> > > [4] Zhao et al. Understanding Byzantine Robustness in Federated Learning with A Black-box Server. arXiv 2024.

---

> ### Author Response · Authors · 2025-11-22
> **Answers to the Reviewer’s Questions (1)**
>
> ### **Q1. Sensitivity to dimension of extracted subspaces**
> We perform an ablation study on the target feature dimension $d$ used in randomized SVD. We vary $d \in \{2048, 1024, 512\}$ and report both performance and runtime:
>
> |ACC (%)|Task-Aware|Task-Agnostic|w/ TIP|
> |-|-|-|-|
> |$d=2048$|85.53|46.07| 84.32|
> |$d=1024$|86.03|47.04| 85.68 |
> |$d=512$|86.26|48.4|86.00|
>
> | FT (%)  | Task-Aware | Task-Agnostic | w/ TIP |
> |-|-|-|-|
> |$d = 2048$|1.41|16.62 | 0.63 |
> |$d = 1024$|1.75|16.55  | 0.10 |
> |$d = 512$|0.96|15.59 | 0.83 |
>
> | Runtime (h) | Base Extract | Total |
> |-|-|-|
> |$d = 2048$|2.26|3.42|
> | $d = 1024$ | 1.45  | 2.62  |
> | $d = 512$ | 1.25   | 2.35  |
>
> Across all settings, both task-aware/agnostic accuracy remain stable (within about $0.7%$), and TIP consistently provides performance gains. Interestingly, the best performance is obtained at $d = 512$, with slightly lower accuracy at $d = 1024, 2048$. We believe that the main reason is noisy or redundant directions - larger $d$ retains low-energy singular components that capture client-specific noise rather than meaningful task structure. Including such directions expands the protected subspace and can unnecessarily constrain learning. In addition, runtime grows significantly with $d$, nearly doubling the Base Extract time from 512 to 2048. Since accuracy remains for most part unaffected, we conclude that choosing $d=512$ represents a good trade-off between efficiency and performance.
>
> ---
> ### **Q2. Optimization instability**
> - **Stability of the projection matrix**: FedProTIP uses an orthogonal projector of the form $P=I-UU^\top$, where $U^\top U=I$. We ensure that $U$ has orthonormal columns via Gram-Schmidt orthogonalization. This prevents accumulation of nearly dependent directions and guarantees that $P$ is a stable, non-expansive projection, i.e., $\lVert Pg \rVert_2 \le \lVert g \rVert_2$ for any gradient $g$. Importantly, FedProTIP treats the global projection matrix as fixed throughout each task; this means that the projector is updated only after a new task shows up, and remains unchanged during local steps and communication rounds. This prevents abrupt switching between projection directions and helps avoid potential instability that could be caused by rapid changes.
> - **Smoothing or Regularization to mitigate oscillation**: When FedProTIP updates a subspace, it uses randomized SVD over activations that are both pooled and randomly sampled across clients and iterations. We keep only the directions whose singular values exceed a threshold, which acts as an implicit smoothing/regularization. In particular, noisy and poorly supported directions end up being discarded, and the resulting projector captures meaningful task-specific variations while avoiding overfitting to outliers.
> ---
> ### **Q3. Local optimizer**
> With standard SGD, FedProTIP local update is $ x_{t,k+1}^i = x_{t,k}^i - \eta_t P_{t-1}\nabla g(x_{t,k}^i)$, where $P_{t-1}$ projects onto the orthogonal complement of the past-tasks subspace. This guarantees that the update direction remains entirely within the plastic subspace $S^\perp$, avoiding by construction interference with previously learned features.
>
> When using adaptive optimizers such as Adam, the update becomes $x_{t, k+1}^i = x_{t,k}^i - \eta_t M_t P_{t-1} \nabla g(x_{t,k}^i)$, where $M_t$ is the diagonal preconditioning matrix. In this case, $M_t$ does not generally commute with $P_{t-1}$, so the product $M_t P_{t-1} \nabla g(x_{t,k}^i)$ may reintroduce components in the protected subspace. This violates the key geometric guarantee of FedProTIP (no updates along past-task directions). In fact, we observe that using Adam leads to performance degradation, supporting the above stated concern. In contrast, plain SGD preserves the projection structure and provides stability. For these reasons, we use SGD in all FedProTIP experiments.
>
> ---
> ### **Q4. Adaptive layer weighting strategy**
>
> In the current version of FedProTIP, we do not use an explicit adaptive layer-weighting strategy. We instead adopt a single global threshold across layers to keep the method simple and avoid introducing additional layer-wise hyperparameters. We will clarify this in the revised manuscript; we view adaptive weighting, such as depth-dependent thresholds or projection scaling, as promising future extensions.
>
> ---
> ### **Q5. Cross-client alignment of subspaces**
>
> FedProTIP aims to provide consistency across clients by extracting activations from the global model rather than client-specific models; the learned feature space is shared across clients. While client heterogeneity affects which samples are seen locally, it does not have a major impact on shared global features. Moreover, recall that at server, the global subspace is formed by aggregating and orthogonalizing local bases – blending clients’ distributions into a unified representation in this way helps avoid misalignment that could take place when using purely local subspace construction.

---

> ### Author Response · Authors · 2025-11-22
> **Answers to the Reviewer’s Questions (2)**
>
> ### **Q6. Privacy leakage from subspace**
> We appreciate the reviewer’s concern regarding potential leakage from the core bases sent to the server. However, FedProTIP is specifically designed to mitigate such risks, both in terms of what information is exposed and how it is processed, and in fact offers significantly more private operational practice than competing methods such as FOT.
>
> First, clients never transmit raw data or full activations. Instead, they extract low-rank orthonormal core bases, i.e., low-rank orthonormal directions, not per-sample feature vectors. The two-stage compression (random subsampling followed by SVD-based projection) results in basis that contains no per-sample vectors or identifying signal-level information
> Second, these bases are then aggregated across many clients and tasks into a single global subspace, further obfuscating individual client contributions and potentially private information. This stands in contrast to FOT, where the server directly centralizes high-dimensional activations that can be exploited in gradient inversion attacks. In FedProTIP, any attempt to reconstruct individual features from the shared subspace $U_l^{(t)}$ is rather challenging since many activation sets share the same low-rank span.
>
> We will further clarify this two-step design (random sampling + local core-basis extraction) in the revised manuscript and explicitly state that FedProTIP centralizes only the compressed, low-dimensional subspace information. We also note that our framework is compatible with additional safeguards (e.g., adding small noise to core bases or secure aggregation of bases) if stronger privacy guarantees are required.
>
> ---
> ### **Q7. Complexity of TIP**
>
> While it is true that the cost of task-identity prediction grows with the number of tasks, in FedProTIP, however, this overhead remains small because (i) TIP uses only the final-layer features and subspaces, and (ii) one can precompute the projection matrices once per task and reuse them for all samples.
>
> We will add the following complexity analysis to the revised manuscript:
> Let $d_L$ denote the final-layer feature dimension, $T$ is the number of tasks, $K$ is the number of clients, and $r_t$ denotes the number of bases for task $t$.
>
> **(i) Precomputation of the projection matrix**: For each task $t$, we precompute the projection matrix $P_t = U_L^{(t)} U_L^{(t)\top}$ once, using the learned bases $U_L^{(t)} \in \mathbb{R}^{d_L \times r_t}$. This costs at most $O(d_L^2 r_t)$ per task and is done offline.
>
> **(ii) Reference vector computation**: For each client, each task $\tau$, and each past task $t \le \tau$, we compute
> $\omega^{(\tau,t)} = \lVert P_t a_L^{(\tau)}\rVert_2$. Given precomputed $P_t$, each projection costs $O(d_L^2)$, so forming all reference vectors on one client costs at most
> $O(T^2 d_L^2)$, and $O(K T^2 d_L^2)$ across $K$ clients. This is a one-time cost during training.
>
> **(iii) TIP at test time**: For test activation $a_L^{\text{te}}$: Computing the relevance vector $\hat{\omega} \in \mathbb{R}^T$ with entries $\hat{\omega}_t = \lVert P_t a_L^{\text{te}}\rVert_2$ requires $T$ projections, yielding the cost of $O(T d_L^2)$. Computing cosine similarities between $\hat{\omega}$ and $T$ stored reference vectors per client costs $O(K T^2)$ (dot products and norms of $T$-dimensional vectors).
>
> In summary, the total TIP overhead per test sample is $O(T d_L^2 + K T^2)$. Since TIP operates only on the final-layer features and $T$ is moderate in our benchmarks, the overhead is negligible compared to the computational costs of backbone inference, which is also reflected in the measured runtimes in Table in our above answer 7 (to the reviewer Z5rx). We'll add per-task runtimes in the manuscript, which shows TIP accounts for less than 1.2% of the total runtime, confirming that task-identity prediction introduces negligible overhead compared to training and GPM subspace updates.
>
>  ---
> ### **Q8. Overlapping tasks**
>
> Following the reviewer’s suggestion, we additionally construct a class-overlapping variant of the benchmark, where 1 or 2 classes are shared between neighboring tasks. The results are reported in the table below. FedProTIP remains robust in this setting: with one overlapping class (10 classes per task / 11 tasks) per task boundary, TIP retains most of its task-aware accuracy. Even when two classes (9 classes per task / 14 tasks) overlap between tasks, TIP still achieves strong task-agnostic performance while maintaining competitive task-aware accuracy and forgetting.
>
> | ACC (%)       | Task-Aware | Task-Agnostic | w/ TIP  |
> |-|-|-|-|
> | **FedProTIP** (1 class) | 84.02 | 42.18  |  83.90 |
> | **FedProTIP** (2 classes) | 82.96 | 28.48 | 76.97 |
>
> | FT (%)       | Task-Aware | Task-Agnostic | w/ TIP  |
> |-|-|-|-|
> | **FedProTIP** (1 class) | 2.41 | 19.86 |  1.43 |
> | **FedProTIP** (2 classes)  | 3.00 | 9.26 | 5.60 |

---

### Official Review · Reviewer_qCdG · 2025-11-01

**Soundness:** 2
**Presentation:** 2
**Contribution:** 2
**Rating:** 4
**Confidence:** 3

**Summary:**

This paper focuses on FedProTIP, a federated continual learning framework for improving the performance on task-agnostic scenario.

The basic idea is twofold: (1) use gradient projection method for mitigating forgetting, and (2) predict the task identity and adjst the global model's outputs accordingly, in order to handle the task-agnostic inference.

**Strengths:**

This paper focuses on an important and timely problem of task-agnostic FCL.

**Weaknesses:**

* The idea of gradient projection is nothing new. There are several related works using the gradient projection idea, as pointed out by the authors.

* It is unclear whether there exist recent works that have proposed some methods for task-agnostic scenario.

* No theoretical guarantee on the success of FedProTIP.

**Questions:**

None

---

> ### Author Response · Authors · 2025-11-21
> **Clarifications About the Contributions and Revisions**
>
> We thank the reviewer for their thoughtful feedback to improve our work. Below, we address each comment in detail:
>
>
> ---
> ### **1. Novelty beyond gradient projection**
> We agree that gradient projection has been explored in prior work on continual learning (e.g., GPM). However, FedProTIP introduces a novel instantiation of this framework tailored for the federated, task-agnostic, and privacy-aware settings as follows:
> -  **Federated gradient projection with subspace aggregation**. Unlike prior methods such as FOT, which project only the aggregated global update and require transmitting full (high-dimensional) activation maps, FedProTIP performs projected local updates on each client using a shared global subspace constructed from much smaller in size and thus communication-efficient core bases. These are aggregated via orthogonalization, thus avoiding raw activation sharing and providing scalability.
> - **Task-identity prediction (TIP) for task-agnostic inference**. Existing projection-based methods, both centralized (GPM) and federated (FOT), assume task-aware inference and require the task ID at test time. In contrast, FedProTIP uses the learned subspaces to enable inference without task ID via an efficient TIP mechanism, without needing generative replay or training additional routers. To our knowledge, this coupling of GPM with task-aware output adjustments has not been explored in FCL literature.
>
> - **Efficiency–performance trade-off in FCL**. FedProTIP is designed to be feasible on edge devices: (i) it employs randomized SVD, (ii) communicates compact orthonormal bases rather than relying on replay buffers or high-dimensional raw activations, and (iii) performs TIP using only final-layer bases. As shown in Section 5.4 and Appendix A.5, it achieves competitive runtime and memory efficiency while delivering state-of-the-art performance across FCL diverse baselines.
>
> Introduction and Related Work sections in our revised manuscript will emphasize these contributions.
>
> ---
> ### **2. Recent task-agnostic continual learning**
> Task-agnostic inference in FCL is relatively underexplored. For a fair and informative benchmarking studies, we implemented federated variants of several recent task-agnostic CL methods that were proposed for centralized settings. These include PEC, SCR, and LODE, please see Table 4 and Section 5.3 for the results. While these methods perform well in centralized scenarios with task-agnostic inference, their federated adaptations suffer from significant accuracy drops under client heterogeneity, consistently underperforming FedProTIP across all benchmarks. We will also include a discussion of a recent federated class-incremental learning method [1] that supports task-agnostic inference via nearest-centroid classification over a unified label space. However, we have been unable to include this method in our empirical study due to the lack of public code at the time of writing, and the non-trivial implementation complexity.
>
> ---
> ### **3. Theoretical guarantee of FedProTIP**
> While our work primarily focuses on empirical effectiveness, FedProTIP is based on sound theoretical principles.
> - **Gradient-projection view and the protection of past tasks**. FedProTIP adopts the classical gradient-projection mechanism where the updates are orthogonally projected to remove components along the subspace spanned by past-task representations. This prevents the model from “unlearning” prior solutions in those directions. A similar motivation underlies the recent CFL method C-FLAG [3], which analyzes forgetting in terms of past-task gradient norms. Our method builds on this intuition by explicitly suppressing destructive updates.
> - **Current task convergence**. The projection only modifies the direction of each local step. In Appendix B, we show that the variance of projected gradients across clients is no larger than in the unprojected case. Hence, standard convergence guarantees for FedAvg (e.g., in convex or smooth nonconvex settings) carry over, since the projected update does not amplify client drift.
> - **Motivation of TIP**. Our main performance gains stem from the TIP. This is consistent with theoretical insights from Kim et al. [2], who show that task-ID prediction quality critically impacts CL performance in task-agnostic settings. FedProTIP combines this with a projection mechanism that is theoretically aligned with existing CFL analyses [3], resulting in a method that both empirically performs well and is conceptually grounded.
>
> In the revised manuscript we will clarify these items and highlight relevant citations.
>
> ---
> [1] Nori et al, Federated Class-Incremental Learning: A Hybrid Approach Using Latent Exemplars and Data-Free Techniques to Address Local and Global Forgetting, ICLR 2025.
>
> [2] Kim et al, A Theoretical Study on Solving Continual Learning, NeurIPS 2022.
>
> [3] Keshri et al, On the Convergence of Continual Federated Learning Using Incrementally Aggregated Gradients, AISTATS 2025.

---

### Meta-Review · Area_Chair_iPA1 · 2026-01-06

**Summary:**

The reviewers shared concerns regarding the algorithmic novelty, the validity of the evaluation setup, and the completeness of the empirical evaluation. The authors' rebuttals addressed the questions on the experiments, and more results were added during the rebuttal period. However, the main concern of the algorithmic novelty is not addressed well. Therefore, I think this paper is slightly below the bar of ICLR, leading to a weak reject decision.

**Reviewer Concerns:**

All the reviewers' concerns about the experiments are mostly addressed.

The algorithmic novelty concern shared by Reviewer qCdG, Z5rx and hsz1 remains outstanding.

**Reviewer Scores:**

Reviewer mHcn might raise his score from 2 to 4, but still slightly negative on this paper.

The other reviewers might not change the scores since the novelty issue has not been well addressed.

---

### Decision · Program_Chairs · 2026-01-26

Reject